# Exposure Bracketing Is All You Need For A High-Quality Image

**Zhilu Zhang, Shuohao Zhang, Renlong Wu, Zifei Yan,**[*] **Wangmeng Zuo**
Harbin Institute of Technology, Harbin, China
`cszlzhang@outlook.com, yhyzshrby@163.com,`
`hirenlongwu@gmail.com, {yanzifei,wmzuo}@hit.edu.cn`

## Abstract

It is highly desired but challenging to acquire high-quality photos with clear content in low-light environments. Although multi-image processing methods (using burst, dual-exposure, or multi-exposure images) have made significant progress in addressing this issue, they typically focus on specific restoration or enhancement problems, and do not fully explore the potential of utilizing multiple images. Motivated by the fact that multi-exposure images are complementary in denoising, deblurring, high dynamic range imaging, and super-resolution, we propose to utilize exposure bracketing photography to get a high-quality image by combining these tasks in this work. Due to the difficulty in collecting real-world pairs, we suggest a solution that first pre-trains the model with synthetic paired data and then adapts it to real-world unlabeled images. In particular, a temporally modulated recurrent network (TMRNet) and self-supervised adaptation method are proposed. Moreover, we construct a data simulation pipeline to synthesize pairs and collect real-world images from 200 nighttime scenarios. Experiments on both datasets show that our method performs favorably against the state-of-the-art multi-image processing ones. Code and datasets are available at `https://github.com/cszhilu1998/BracketIRE`.

## 1 Introduction

In low-light environments, capturing visually appealing photos with clear content presents a highly desirable yet challenging goal. When adopting a low exposure time, the camera only captures a small amount of photons, introducing inevitable noise and rendering dark areas invisible. When taking a high exposure time, camera shake and object movement result in blurry images, in which bright areas may be overexposed. Although single-image restoration (*e.g.*, denoising (Zhang et al., 2017; 2018b; Guo et al., 2019; Brooks et al., 2019; Zamir et al., 2020; Abdelhamed et al., 2020; Li et al., 2023), deblurring (Nah et al., 2017; Zhang et al., 2018a; Tao et al., 2018; Cho et al., 2021; Zamir et al., 2021; Mao et al., 2023), and super-resolution (SR) (Dong et al., 2015; Lim et al., 2017; Zhang et al., 2018e;c; Liu et al., 2020a; Liang et al., 2021; Ledig et al., 2017)) and enhancement (*e.g.*, high dynamic range (HDR) reconstruction (Eilertsen et al., 2017; Liu et al., 2020b; Zou et al., 2023; Pérez-Pellitero et al., 2021; Lee et al., 2018; Chen et al., 2023)) methods have been extensively investigated, their performance is constrained by the severely ill-posed problems.

Recently, leveraging multiple images for image restoration and enhancement has demonstrated potential in addressing this issue, thereby attracting increasing attention. We provide a summary of several related settings and methods in Tab. 1. For example, some burst image restoration methods (Bhat et al., 2021a;b; Dudhane et al., 2022; Luo et al., 2022; Lecouat et al., 2021; Mehta et al., 2023; Dudhane et al., 2023; Bhat et al., 2023; Wu et al., 2023; Bhat et al., 2022) utilize multiple consecutive frames with the same exposure time as inputs, being able to perform SR and denoising. The works based on dual-exposure images (Yuan et al., 2007; Chang et al., 2021; Mustaniemi et al., 2020; Zhao et al., 2022; Zhang et al., 2022b; Shekarforoush et al., 2023; Lai et al., 2022) combine the short-exposure noisy and long-exposure blurry pairs for better restoration. Multi-exposure images are commonly employed for HDR imaging (Kalantari et al., 2017; Yan et al., 2019; Prabhakar et al.,

---

[*]Corresponding Author.

Table 1: Comparison between various multi-image processing manners

| Setting | Methods | Input Images | Supported Tasks | | | |
|---|---|---|---|---|---|---|
| | | | Denoising | Deblurring | HDR | SR |
| Burst Denoising | (Godard et al., 2018; Xia et al., 2020; Rong et al., 2020; Guo et al., 2022) | Burst | ✓ | | | |
| Burst Deblurring | (Wieschollek et al., 2017; Peña et al., 2019; Aittala & Durand, 2018) | Burst | | ✓ | | |
| Burst SR | (Deudon et al., 2020; Wronski et al., 2019; Wei et al., 2023) | Burst | | | | ✓ |
| Burst Denoising and SR | (Bhat et al., 2021a;b; Dudhane et al., 2022; Luo et al., 2022; Lecouat et al., 2021; Mehta et al., 2023; Dudhane et al., 2023; Bhat et al., 2023; Wu et al., 2023; Bhat et al., 2022) | Burst | ✓ | | | ✓ |
| Burst Denoising and HDR | (Hasinoff et al., 2016; Ernst & Wronski, 2021) | Burst | ✓ | | ✓ | |
| Dual-Exposure Image Restoration | (Chang et al., 2021; Mustaniemi et al., 2020; Zhao et al., 2022; Zhang et al., 2022b; Shekarforoush et al., 2023; Lai et al., 2022) | Dual-Exposure | ✓ | ✓ | | |
| Basic HDR Imaging | (Kalantari et al., 2017; Yan et al., 2019; Niu et al., 2021; Liu et al., 2022; Yan et al., 2023a; Tel et al., 2023; Zhang et al., 2024b) | Multi-Exposure | | | ✓ | |
| HDR Imaging with Denoising | (Hasinoff et al., 2010; Liu et al., 2023; Chi et al., 2023; Pérez-Pellitero et al., 2021) | Multi-Exposure | ✓ | | ✓ | |
| HDR Imaging with SR | (Tan et al., 2021) | Multi-Exposure | | | ✓ | ✓ |
| HDR Imaging with Denoising and SR | (Lecouat et al., 2022) | Multi-Exposure | ✓ | | ✓ | ✓ |
| Our BracketIRE | - | Multi-Exposure | ✓ | ✓ | ✓ | |
| Our BracketIRE+ | - | Multi-Exposure | ✓ | ✓ | ✓ | ✓ |

2019; Wu et al., 2018; Niu et al., 2021; Liu et al., 2022; Yan et al., 2023a; Tel et al., 2023; Zhang et al., 2024b; Song et al., 2022).

Nevertheless, in night scenarios, it remains unfeasible to obtain noise-free, blur-free, and HDR images when employing these multi-image processing methods. On the one hand, burst and dual-exposure images both possess restricted dynamic ranges, constraining the potential expansion of the two manners into HDR reconstruction. On the other hand, most HDR reconstruction approaches based on multi-exposure images are constructed with the ideal assumption that image noise and blur are not taken into account, which results in their inability to restore degraded images. Although recent works (Liu et al., 2023; Chi et al., 2023; Lecouat et al., 2022) have combined with denoising task, blur in long-exposure images has not been incorporated into them, which is still inconsistent with real-world multi-exposure images.

In fact, considering all multi-exposure factors (including noise, blur, underexposure, overexposure, and misalignment) is not only beneficial to practical applications, but also offers us an opportunity to combine image restoration and enhancement tasks to get a high-quality image. **First**, the independence and randomness of noise (Wei et al., 2020) between images allow them to assist each other in denoising, and its motivation is similar to that of burst denoising (Mildenhall et al., 2018; Godard et al., 2018; Xia et al., 2020; Rong et al., 2020; Guo et al., 2022). In particular, as demonstrated in dual-exposure restoration works (Yuan et al., 2007; Chang et al., 2021; Mustaniemi et al., 2020; Zhao et al., 2022; Zhang et al., 2022b; Shekarforoush et al., 2023; Lai et al., 2022), long-exposure images with a higher signal-to-noise ratio can play a significantly positive role in removing noise from the short-exposure images. **Second**, the shortest-exposure image can be considered blur-free. It can offer sharp guidance for deblurring longer-exposure images. **Third**, underexposed areas in the short-exposure image may be well-exposed in the long-exposure one, while overexposed regions in the long-exposure image may be clear in the short-exposure one. Combining multi-exposure images makes HDR imaging easier than single-image enhancement. **Fourth**, the sub-pixel shift between multiple images caused by camera shake or motion is conducive to multi-frame SR (Wronski et al., 2019). In summary, leveraging the complementarity of multi-exposure images offers the potential to integrate the four problems (*i.e.*, denoising, deblurring, HDR reconstruction, and SR) into a unified framework that can generate a noise-free, blur-free, high dynamic range, and high-resolution image.

Specifically, in terms of tasks, we first utilize bracketing photography to combine basic restoration (*i.e.*, denoising and deblurring) and enhancement (*i.e.*, HDR reconstruction), named BracketIRE. Then we append the SR task, dubbed BracketIRE+, as shown in Tab. 1. In terms of methods, due to the difficulty of collecting real-world paired data, we achieve that through supervised pre-training on synthetic pairs and self-supervised adaptation on real-world images. On the one hand, we adopt the recurrent network manner as the basic framework, which is inspired by its successful applications in processing sequence images, *e.g.*, burst (Guo et al., 2022; Rong et al., 2020; Wu et al., 2023)

and video (Wang et al., 2023b; Chan et al., 2021; 2022) restoration. Nevertheless, sharing the same restoration parameters for each frame may result in limited performance, as degradations (*e.g.*, blur, noise, and color) vary between different multi-exposure images. To alleviate this problem, we propose a temporally modulated recurrent network (TMRNet), where each frame not only shares some parameters with others, but also has its own specific ones. On the other hand, pre-trained TMRNet on synthetic data has limited generalization ability and sometimes produces unpleasant artifacts in the real world, due to the inevitable gap between simulated and real images. For that, we propose a self-supervised adaptation method. In particular, we utilize the temporal characteristics of multi-exposure image processing to design learning objectives to fine-tune TMRNet.

For training and evaluation, we construct a pipeline for synthesizing data pairs, and collect real-world images from 200 nighttime scenarios with a smartphone. The two datasets also provide benchmarks for future studies. We conduct extensive experiments, which show that the proposed method achieves state-of-the-art performance in comparison with other multi-image processing ones.

The contributions can be summarized as follows:

- We propose to utilize exposure bracketing photography to get a high-quality (*i.e.*, noise-free, blur-free, high dynamic range, and high-resolution) image by combining image denoising, deblurring, high dynamic range reconstruction, and super-resolution tasks.
- We suggest a solution that first pre-trains the model with synthetic pairs and then adapts it to unlabeled real-world images, where a temporally modulated recurrent network and a self-supervised adaptation method are proposed.
- Experiments on both synthetic and captured real-world datasets show the proposed method outperforms the state-of-the-art multi-image processing ones.

## 2 RELATED WORK

### 2.1 SUPERVISED MULTI-IMAGE PROCESSING.

**Burst Image Restoration and Enhancement.** Burst-based manners generally leverage multiple consecutive frames with the same exposure for image processing. Most methods focus on image restoration, such as denoising, deblurring, and SR tasks, as shown in Tab. 1. And they mainly explore inter-frame alignment and feature fusion manners. The former can be implemented by utilizing various techniques, *e.g.*, homography transformation (Wei et al., 2023), optical flow (Ranjan & Black, 2017; Bhat et al., 2021a;b), deformable convolution (Dai et al., 2017; Luo et al., 2022; Dudhane et al., 2023; Guo et al., 2022), and cross-attention (Mehta et al., 2023). The latter are also developed with multiple routes, *e.g.*, weighted-based mechanism (Bhat et al., 2021a;b), kernel predition (Xia et al., 2020; Mildenhall et al., 2018; Dahary et al., 2021), attention-based merging (Dudhane et al., 2023; Mehta et al., 2023), and recursive fusion (Deudon et al., 2020; Guo et al., 2022; Rong et al., 2020; Wu et al., 2023). Moreover, HDR+ (Hasinoff et al., 2016) joins HDR imaging and denoising by capturing underexposure raw bursts. Recent updates (Ernst & Wronski, 2021) of HDR+ introduce additional well-exposed frames for improving performance. Although such manners may be suitable for scenes with moderate dynamic range, they have limited ability for scenes with high dynamic range.

**Dual-Exposure Image Restoration.** Several methods (Yuan et al., 2007; Chang et al., 2021; Mustaniemi et al., 2020; Zhao et al., 2022; Zhang et al., 2022b; Shekarforoush et al., 2023; Lai et al., 2022) exploit the complementarity of short-exposure noisy and long-exposure blurry images for better restoration. For example, Yuan *et al.* (Yuan et al., 2007) estimates blur kernels by exploring the texture of short-exposure images and then employ the kernels to deblur long-exposure ones. Mustaniemi *et al.* (Mustaniemi et al., 2020) and Chang *et al.* (Chang et al., 2021) deploy convolutional neural networks (CNN) to aggregate dual-exposure images, achieving superior results compared with single-image methods on synthetic data. D2HNet (Zhao et al., 2022) proposes a two-phase DeblurNet-EnhanceNet architecture for real-world image restoration. However, few works join it with HDR imaging, mainly due to the restricted dynamic range of dual-exposure images.

**Multi-Exposure HDR Image Reconstruction.** Multi-exposure images are widely used for HDR image reconstruction. Most methods (Kalantari et al., 2017; Yan et al., 2019; Prabhakar et al., 2019; Wu et al., 2018; Niu et al., 2021; Liu et al., 2022; Yan et al., 2023a; Tel et al., 2023; Zhang et al., 2024b; Song et al., 2022) only focus on removing ghosting caused by image misalignment. For

instance, Kalantari (Kalantari et al., 2017) align multi-exposure images and then propose a data-driven approach to merge them. AHDRNet (Yan et al., 2019) utilizes spatial attention and dilated convolution to achieve deghosting. HDR-Transformer (Liu et al., 2022) and SCTNet (Tel et al., 2023) introduce self-attention and cross-attention to enhance feature interaction, respectively. Besides, a few methods (Hasinoff et al., 2010; Liu et al., 2023; Chi et al., 2023; Pérez-Pellitero et al., 2021) take noise into account. Kim *et al*. (Kim & Kim, 2023) further introduce motion blur in the long-exposure image. However, the unrealistic blur simulation approach and the requirements of time-varying exposure sensors limit its practical applications. In this work, we consider more realistic situations in low-light environments, and incorporate both severe noise and blur. More importantly, we propose to utilize the complementary potential of multi-exposure images to combine image restoration and enhancement tasks, including image denoising, deblurring, HDR reconstruction, and SR.

## 2.2 SELF-SUPERVISED MULTI-IMAGE PROCESSING

The complementarity of multiple images enables the achievement of certain image processing tasks in a self-supervised manner. For self-supervised image restoration, some works (Dewil et al., 2021; Ehret et al., 2019; Sheth et al., 2021; Wang et al., 2023c) accomplish multi-frame denoising with the assistance of Noise2Noise (Lehtinen et al., 2018) or blind-spot networks (Laine et al., 2019; Wu et al., 2020; Krull et al., 2019). SelfIR (Zhang et al., 2022b) employs a collaborative learning framework for restoring noisy and blurry images. Bhat *et al*. (Bhat et al., 2023) propose self-supervised Burst SR by establishing a reconstruction objective that models the relationship between the noisy burst and the clean image. Self-supervised real-world SR can also be addressed by combining short-focus and telephoto images (Zhang et al., 2022a; Wang et al., 2021; Xu et al., 2023). For self-supervised HDR reconstruction, several works (Prabhakar et al., 2021; Yan et al., 2023b; Nazarczuk et al., 2022) generate or search pseudo-pairs for training the model, while SelfHDR (Zhang et al., 2024b) decomposes the potential GT into constructable color and structure supervision. However, these methods can only handle specific degradations, making them less practical for our task with multiple ones. In this work, instead of creating self-supervised algorithms trained from scratch, we suggest adapting the model trained on synthetic pairs to real images, and utilize the temporal characteristics of multi-exposure image processing to design self-supervised learning objectives.

## 3 METHOD

### 3.1 PROBLEM DEFINITION AND FORMULATION

Denote the scene irradiance at time $t$ by $\mathbf{X}(t)$. When capturing a raw image $\mathbf{Y}$ at $t_0$ time, we can simplify the camera's image formation model as,

$$\mathbf{Y} = \mathcal{M}(\int_{t_0}^{t_0+\Delta t} \mathcal{D}(\mathcal{W}_t(\mathbf{X}(t)))d\boldsymbol{t} + \mathbf{N}).$$ (1)

In this equation, (1) $\mathcal{D}$ is a spatial sampling function, which is mainly related to sensor size. This function limits the image resolution. (2) $\Delta t$ denotes exposure time and $\mathcal{W}_t$ represents the warp operation that accounts for camera shake. Combined with potential object movements in $\mathbf{X}(t)$, the integral formula $\int$ can result in a blurry image, especially when $\Delta t$ is long (Nah et al., 2017). (3) $\mathbf{N}$ represents the inevitable noise, *e.g.*, read and shot noise (Brooks et al., 2019). (4) $\mathcal{M}$ maps the signal to integer values ranging from 0 to $2^b - 1$, where $b$ denotes the bit depth of the sensor. This mapping may reduce the dynamic range of the scene (Lecouat et al., 2022). In summary, the imaging process introduces multiple degradations, including blur, noise, as well as a decrease in dynamic range and resolution. Notably, in low-light conditions, some degradations (*e.g.*, noise) may be more severe.

In pursuit of higher-quality images, substantial efforts have been made to deal with the inverse problem through single-image or multi-image restoration (*e.g.*, denoising, deblurring, and SR) and enhancement (*e.g.*, HDR imaging). However, most efforts tend to focus on addressing partial degradations, and few works encompass all these aspects, as shown in Tab. 1. In this work, inspired by the complementary potential of multi-exposure images, we propose to exploit bracketing photography to integrate these tasks for noise-free, blur-free, high dynamic range, and high-resolution images.

Specifically, the proposed BracketIRE involves denoising, deblurring, and HDR reconstruction, while BracketIRE+ adds support for SR task. Here, we provide a formalization for them. Firstly, We define

the number of input multi-exposure images as $T$, and define the raw image taken with exposure time $\Delta t_i$ as $\mathbf{Y}_i$, where $i \in \{1, 2, ..., T\}$ and $\Delta t_i < \Delta t_{i+1}$. Then, we follows the recommendations from multi-exposure HDR reconstruction methods (Yan et al., 2019; Niu et al., 2021; Liu et al., 2022; Yan et al., 2023a; Tel et al., 2023), normalizing $\mathbf{Y}_i$ to $\frac{\mathbf{Y}_i}{\Delta t_i / \Delta t_1}$ and concatenating it with its gamma-transformed image, *i.e.*,

$$\mathbf{Y}_i^c = \{\frac{\mathbf{Y}_i}{\Delta t_i / \Delta t_1}, (\frac{\mathbf{Y}_i}{\Delta t_i / \Delta t_1})^\gamma\}, \tag{2}$$

where $\gamma$ represents the gamma correction parameter and is generally set to $1/2.2$. Finally, we feed these concatenated images into BracketIRE or BracketIRE+ model $\mathcal{B}$ with parameters $\Theta_\mathcal{B}$, *i.e.*,

$$\hat{\mathbf{X}} = \mathcal{B}(\{\mathbf{Y}_i^c\}_{i=1}^T; \Theta_\mathcal{B}), \tag{3}$$

where $\hat{\mathbf{X}}$ is the generated image. Furthermore, the optimized network parameters can be written as,

$$\Theta_\mathcal{B}^* = \arg\min_{\Theta_\mathcal{B}} \mathcal{L}_\mathcal{B}(\mathcal{T}(\hat{\mathbf{X}}), \mathcal{T}(\mathbf{X})), \tag{4}$$

where $\mathcal{L}_\mathcal{B}$ represents the loss function, and can adopt $\ell_1$ loss. $\mathbf{X}$ is the ground-truth (GT) image. $\mathcal{T}(\cdot)$ denotes the $\mu$-law based tone-mapping operator (Kalantari et al., 2017), *i.e.*,

$$\mathcal{T}(\mathbf{X}) = \frac{\log(1 + \mu \mathbf{X})}{\log(1 + \mu)}, \text{ where } \mu = 5,000. \tag{5}$$

Besides, we consider the shortest-exposure image (*i.e.*, $\mathbf{Y}_1$) blur-free and take it as a spatial alignment reference for other frames. In other words, the output $\hat{\mathbf{X}}$ should be aligned strictly with $\mathbf{Y}_1$.

Towards real-world dynamic scenarios, it is nearly impossible to capture GT $\mathbf{X}$, and it is hard to develop self-supervised algorithms trained on real-world images from scratch. To address the issue, we suggest pre-training the model on synthetic pairs first and then adapting it to real-world scenarios in a self-supervised manner. In particular, we propose a temporally modulated recurrent network for BracketIRE and BracketIRE+ tasks in Sec. 3.2, and a self-supervised adaptation method in Sec. 3.3.

## 3.2 TEMPORALLY MODULATED RECURRENT NETWORK

Recurrent networks have been successfully applied to burst (Wu et al., 2023) and video (Wang et al., 2023b; Chan et al., 2021; 2022) restoration methods, which generally involve four modules, *i.e.*, feature extraction, alignment, aggregation, and reconstruction module. Here we adopt a unidirectional recurrent network as our baseline, and briefly describe its pipeline. Firstly, the multi-exposure images $\{\mathbf{Y}_i^c\}_{i=1}^T$ are fed into an encoder for extracting features $\{\mathbf{F}_i\}_{i=1}^T$. Then, the alignment module is deployed to align $\mathbf{F}_i$ with reference feature $\mathbf{F}_1$, getting the aligned feature $\tilde{\mathbf{F}}_i$. Next, the aggregation module $\mathcal{A}$ takes $\tilde{\mathbf{F}}_i$ and the previous temporal feature $\mathbf{H}_{i-1}$ as inputs, generating the current fused feature $\mathbf{H}_i$, *i.e.*,

$$\mathbf{H}_i = \mathcal{A}(\tilde{\mathbf{F}}_i, \mathbf{H}_{i-1}; \Theta_\mathcal{A}), \tag{6}$$

where $\Theta_\mathcal{A}$ denotes the parameters of $\mathcal{A}$. Finally, $\mathbf{H}_T$ is fed into the reconstruction module to output the result.

The aggregation module plays a crucial role in the recurrent framework and usually takes up most of the parameters. In burst and video restoration tasks, the degradation types of multiple input frames are generally the same, so it is appropriate for frames to share the same aggregation network parameters $\Theta_\mathcal{A}$. In BracketIRE and BracketIRE+ tasks, the noise models of multi-exposure images may be similar, as they can be taken by the same device. However, other degradations are varying. For example, the longer the exposure time, the more serious the image blur, the fewer underexposed areas, and the more overexposed ones. Thus, sharing $\Theta_\mathcal{A}$ may limit performance.

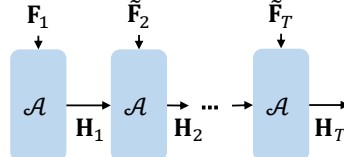

(a) Baseline Recurrent Network

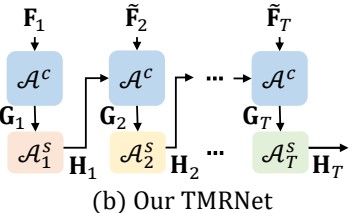

(b) Our TMRNet

Figure 1: Illustration of baseline recurrent network (*e.g.*, RBSR (Wu et al., 2023)) and our TMRNet. Instead of sharing parameters of aggregation module $\mathcal{A}$ for all frames, we divide it into a common one $\mathcal{A}^c$ for all frames and a specific one $\mathcal{A}_i^s$ only for $i$-th frame. Modules with different colors have different parameters.

(a) Temporally Self-Supervised Loss     (b) EMA Regularization Loss

Figure 2: Self-supervised loss terms for real-image adaptation. $\mathcal{B}$ denotes TMRNet for BracketIRE or BracketIRE+ task. In sub-figure (a), an integer from 1 to $R$ $(R < T)$ is randomly chosen as $r$. In sub-figure (b), EMA denotes exponential moving average.

To alleviate this problem, we suggest assigning specific parameters for each frame while sharing some ones, thus proposing a temporally modulated recurrent network (TMRNet). As shown in Fig. 1, we divide the aggregation module $\mathcal{A}$ into a common one $\mathcal{A}^c$ for all frames and a specific one $\mathcal{A}_i^s$ only for $i$-th frame. Features are first processed via $\mathcal{A}^c$ and then further modulated via $\mathcal{A}_i^s$. Eq. (6) can be modified as,

$$\begin{aligned} \mathbf{G}_i &= \mathcal{A}^c(\tilde{\mathbf{F}}_i, \mathbf{H}_{i-1}; \Theta_{\mathcal{A}^c}), \\ \mathbf{H}_i &= \mathcal{A}_i^s(\mathbf{G}_i; \Theta_{\mathcal{A}_i^s}), \end{aligned} \qquad (7)$$

where $\mathbf{G}_i$ represents intermediate features, $\Theta_{\mathcal{A}^c}$ and $\Theta_{\mathcal{A}_i^s}$ denote the parameters of $\mathcal{A}^c$ and $\mathcal{A}_i^s$, respectively. We do not design complex architectures for $\mathcal{A}^c$ and $\mathcal{A}_i^s$, and each one only consists of a 3×3 convolution layer followed by some residual blocks (He et al., 2016). More details of TMRNet can be seen in Sec. 5.1.

### 3.3 SELF-SUPERVISED REAL-IMAGE ADAPTATION

It is hard to simulate multi-exposure images with diverse variables (*e.g.*, noise, blur, brightness, and movement) that are completely consistent with real-world ones. Due to the inevitable gap, models trained on synthetic pairs have limited generalization capabilities in real scenarios. Undesirable artifacts are sometimes produced and some details are missed. To address the issue, we propose to perform self-supervised adaptation for real-world unlabeled images.

Specifically, we explore the temporal characteristics of multi-exposure image processing to design self-supervised loss terms elaborately, as shown in Fig. 2. Denote the model output of inputting the previous $r$ frames $\{\mathbf{Y}_i^c\}_{i=1}^r$ by $\hat{\mathbf{X}}_r$. Generally, $\hat{\mathbf{X}}_T$ performs better than $\hat{\mathbf{X}}_r$ $(r < T)$, as shown in Sec. 6.1. For supervising $\hat{\mathbf{X}}_r$, although no ground-truth is provided, $\hat{\mathbf{X}}_T$ can be taken as the pseudo-target. Thus, the temporally self-supervised loss can be written as,

$$\mathcal{L}_{self} = ||\mathcal{T}(\hat{\mathbf{X}}_r) - \mathcal{T}(sg(\hat{\mathbf{X}}_T))||_1, \qquad (8)$$

where $r$ is randomly selected from 1 to $R$ $(R < T)$, $sg(\cdot)$ denotes the stop-gradient operator.

Nevertheless, only deploying $\mathcal{L}_{self}$ can easily lead to trivial solutions, as the final output $\hat{\mathbf{X}}_T$ is not subject to any constraints. To stabilize training process, we suggest an exponential moving average (EMA) regularization loss, which constrains the output $\hat{\mathbf{X}}_T$ of the current iteration to be not too far away from that of previous ones. It can be written as,

$$\mathcal{L}_{ema} = ||\mathcal{T}(\hat{\mathbf{X}}_T) - \mathcal{T}(sg(\hat{\mathbf{X}}_T^{ema}))||_1, \qquad (9)$$

where $\hat{\mathbf{X}}_T^{ema} = \mathcal{B}(\{\mathbf{Y}_i^c\}_{i=1}^T; \Theta_{\mathcal{B}}^{ema})$ and $\Theta_{\mathcal{B}}^{ema}$ denotes EMA parameters in the current iteration. Denote model parameters in the $k$-th iteration by $\Theta_{\mathcal{B}_k}$, the EMA parameters in the $k$-th iteration can be written as,

$$\Theta_{\mathcal{B}_k}^{ema} = a\Theta_{\mathcal{B}_{k-1}}^{ema} + (1-a)\Theta_{\mathcal{B}_k}, \qquad (10)$$

where $\Theta_{\mathcal{B}_0}^{ema} = \Theta_{\mathcal{B}_0}$ and $a = 0.999$.

The total adaptation loss is the combination of $\mathcal{L}_{ema}$ and $\mathcal{L}_{self}$, *i.e.*,

$$\mathcal{L}_{ada} = \mathcal{L}_{ema} + \lambda_{self}\mathcal{L}_{self}, \qquad (11)$$

where $\lambda_{self}$ is the weight of $\mathcal{L}_{self}$.

## 4 DATASETS

### 4.1 SYNTHETIC PAIRED DATASET

Although it is unrealistic to synthesize perfect multi-exposure images, we should still shorten the gap with the real images as much as possible. In the camera's imaging model in Eq. (1), noise, blur, motion, and dynamic range of multi-exposure images should be carefully designed.

Video provides a better basis than a single image in simulating motion and blur of multi-exposure images. We start with HDR videos from Froehlich *et al.* (Froehlich et al., 2014)[1] to construct the simulation pipeline. First, we follow the suggestion from Nah *et al.* (Nah et al., 2019) to perform frame interpolation, as these low frame rate (∼25 fps) videos are unsuitable for synthesizing blur. RIFE (Huang et al., 2022) is adopted for increasing the frame rate by 32 times. Then, we convert these RGB videos to raw space with Bayer pattern according to UPI (Brooks et al., 2019), getting HDR raw sequences $\{\mathbf{V}_m\}_{m=1}^M$. The first frame $\mathbf{V}_1$ is taken as a GT.

Next, we utilize $\{\mathbf{V}_m\}_{m=1}^M$ and introduce degradations to construct multi-exposure images. The process mainly includes the following 5 steps. (1) Bicubic $4\times$ down-sampling is applied to obtain low-resolution images, which is optional and serves for BracketIRE+ task. (2) The video is split into $T$ non-overlapped groups, where $i$-th group should be used to synthesize $\mathbf{Y}_i$. Such grouping utilizes the motion in the video itself to simulate motion between $T$ multi-exposure images. (3) Denote the exposure time ratio between $\mathbf{Y}_i$ and $\mathbf{Y}_{i-1}$ by $S$. We sequentially move $S^{i-1}$ ($\{i-1\}$-th power of $S$) consecutive images into the above $i$-th group, and sum them up to simulate blurry images. (4) We transform the HDR blurry images into low dynamic range (LDR) ones by cropping values outside the specified range and mapping the cropped values to 10-bit unsigned integers. (5) We add the heteroscedastic Gaussian noise (Brooks et al., 2019; Wang et al., 2020; Hasinoff et al., 2010) to LDR images to generate the final multi-exposure images (*i.e.*, $\{\mathbf{Y}_i\}_{i=1}^T$). The noise variance is a function of pixel intensity, whose parameters are estimated from the captured real-world images in Sec. 4.2. More details of RGB-to-RAW conversion, frame interpolation, and noise can be seen in Appendix A.1, Appendix A.2, and Appendix A.3, respectively.

Besides, we set the exposure time ratio $S$ to 4 and the frame number $T$ to 5, as it can cover most of the dynamic range with fewer images. The GT has a resolution of 1,920×1,080 pixels. Finally, we obtain 1,335 data pairs from 35 scenes. 1,045 pairs from 31 scenes are used for training, and the remaining 290 pairs from the other 4 scenes are used for testing.

### 4.2 REAL-WORLD DATASET

Real-world multi-exposure images are collected with the main camera of Xiaomi 10S smartphone at night. Specifically, we utilize the bracketing photography function in ProShot (Games, 2023) application (APP) to capture raw images with a resolution of 6,016×4,512 pixels. The exposure time ratio $S$ is set to 4, the frame number $T$ is set to 5, ISO is set to 1,600; these values are also the maximum available settings in APP. The exposure time of the medium-exposure image (*i.e.*, $\mathbf{Y}_3$) is automatically adjusted by APP. Thus, other exposures can be obtained based on $S$. It is worth noting that we hold the smartphone for shooting, without any stabilizing device, which aims to bring in the realistic hand-held shake. Besides, both static and dynamic scenes are collected, with a total of 200. 100 scenes are used for training and the other 100 are used for evaluation.

## 5 EXPERIMENTS

### 5.1 IMPLEMENTATION DETAILS

**Network Details.** The input multi-exposure images and ground-truth HDR image are both 4-channel data packed from raw images with the Bayer pattern. Following settings in RBSR (Wu et al., 2023), the encoder and reconstruction module consist of 5 residual blocks (He et al., 2016), the alignment module adopts flow-guided deformable approach (Chan et al., 2022). Besides, the total number of residual blocks in aggregation module remains the same as that of RBSR (Wu et al., 2023), *i.e.*, 40,

---

[1]The dataset is licensed under CC BY and is publicly available at the site.

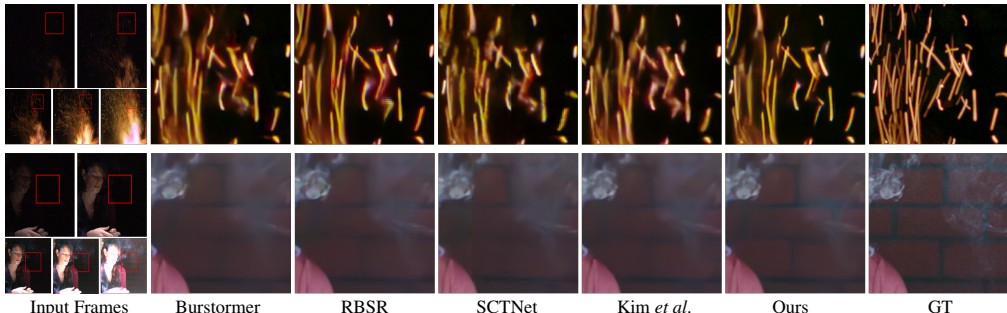

| Input Frames | Burstormer | RBSR | SCTNet | Kim *et al.* | Ours | GT |

Figure 3: Visual comparison on the synthetic dataset of BracketIRE task. Our method restores sharper edges and clearer details.

where the common module has 16 and the specific one has 24. For BracketIRE+ task, we additionally deploy PixelShuffle (Shi et al., 2016) at the end of networks for up-sampling features.

**Training Details.** We randomly crop patches and augment them with flips and rotations. The batch size is set to 8. The input patch size is $128 \times 128$ and $64 \times 64$ for BracketIRE and BracketIRE+ tasks, respectively. We adopt AdamW (Loshchilov & Hutter, 2017) optimizer with $\beta_1 = 0.9$ and $\beta_2 = 0.999$. Models are trained for 400 epochs ($\sim 60$ hours) on synthetic images and fine-tuned for 10 epochs ($\sim 2.6$ hours) on real-world ones, with the initial learning rate of $10^{-4}$ and $7.5 \times 10^{-5}$, respectively. Cosine annealing strategy (Loshchilov & Hutter, 2016) is employed to decrease the learning rates to $10^{-6}$. $r$ is randomly selected from $\{1, 2, 3\}$. $\lambda_{self}$ is set to 1. Moreover, BracketIRE+ models are initialized with pre-trained BracketIRE models on synthetic experiments. All experiments are conducted using PyTorch (Paszke et al., 2019) on a single Nvidia RTX A6000 (48GB) GPU.

## 5.2 EVALUATION AND COMPARISON CONFIGURATIONS

**Evaluation Configurations.** For quantitative evaluations and visualizations, we first convert raw results to linear RGB space through a post-processing pipeline and then tone-map them with Eq. (5), getting 16-bit RGB images. All metrics are computed on the RGB images. For synthetic experiments, we adopt PSNR, SSIM (Wang et al., 2004), and LPIPS (Zhang et al., 2018d) metrics. 10 and 4 invalid pixels around the original input image are excluded for BracketIRE and BracketIRE+ tasks, respectively. Kindly refer to Appendix A.4 for the reason. For real-world ones, we employ no-reference metrics *i.e.*, CLIPIQA (Wang et al., 2023a) and MANIQA (Yang et al., 2022).

**Comparison Configurations.** We compare the proposed method with 10 related state-of-the-art networks, including 5 burst processing ones (*i.e.*, DBSR (Bhat et al., 2021a), MFIR (Bhat et al., 2021b), BIPNet (Dudhane et al., 2022), Burstormer (Dudhane et al., 2023) and RBSR (Wu et al., 2023)) and 5 HDR reconstruction ones (*i.e.*, AHDRNet (Yan et al., 2019), HDRGAN (Niu et al., 2021), HDR-Tran. (Liu et al., 2022), SCTNet (Tel et al., 2023) and Kim *et al.* (Kim & Kim, 2023)). For a fair comparison, we modify their models to adapt inputs with 5 frames, and retrain them on our synthetic pairs following the formulation in Sec. 3.1. When testing real-world images, their trained models are deployed directly, while our models are fine-tuned on real-world training images with the proposed self-supervised adaptation method.

## 5.3 EXPERIMENTAL RESULTS

**Results on Synthetic Dataset.** We summarize the quantitative results in Tab. 2. On BracketIRE task, we achieve 0.25dB and 0.26dB PSNR gains than RBSR (Wu et al., 2023) and Kim *et al.* (Kim & Kim, 2023), respectively, which are the latest state-of-the-art methods. On BracketIRE+ task, the improvements are 0.16dB and 0.37dB, respectively. It demonstrates the effectiveness of our TMRNet, which handles the varying degradations of multi-exposure images by deploying frame-specific parameters. Moreover, the qualitative results in Fig. 3 show that TMRNet recovers more realistic details than others.

**Results on Real-World Dataset.** We achieve the best no-reference scores on BracketIRE task and the highest CLIPIQA (Wang et al., 2023a) on BracketIRE+ task. But note that the no-reference

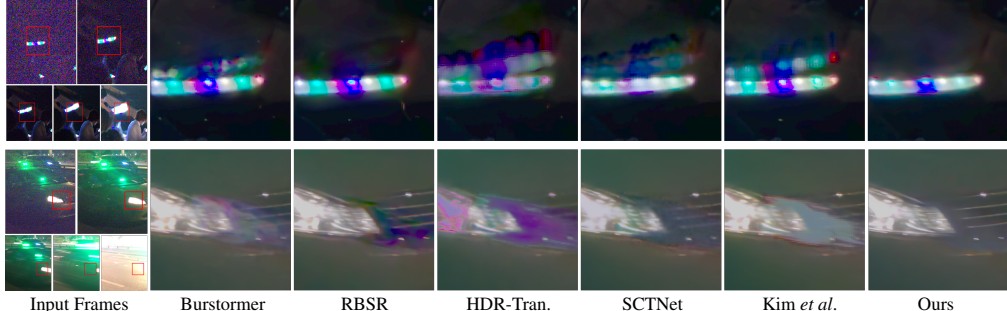

| Input Frames | Burstormer | RBSR | HDR-Tran. | SCTNet | Kim *et al.* | Ours |

Figure 4: Visual comparison on the real-world dataset of BracketIRE task. Note that there is no ground-truth. Our results have fewer ghosting artifacts.

Table 2: Quantitative comparison with state-of-the-art methods on the synthetic and real-world datasets of BracketIRE and BracketIRE+ tasks, respectively. 'Ada.' means the self-supervised real-image adaptation. The top two results are marked in **bold** and underlined, respectively.

| | | BracketIRE | | BracketIRE+ | |
| | Method | Synthetic | Real-World | Synthetic | Real-World |
| | | PSNR↑/SSIM↑/LPIPS↓ | CLIPIQA↑/MANIQA↑ | PSNR↑/SSIM↑/LPIPS↓ | CLIPIQA↑/MANIQA↑ |
|---|---|---|---|---|---|
| | DBSR | 35.13/0.9092/0.188 | 0.1359/0.1653 | 29.79/0.8546/0.335 | 0.3340/0.2911 |
| Burst | MFIR | 35.64/0.9161/0.177 | 0.2192/0.2310 | 30.06/0.8591/0.319 | 0.3402/0.2908 |
| Processing | BIPNet | 36.92/0.9331/0.148 | 0.2234/0.2348 | 30.02/0.8582/0.324 | 0.3577/0.2979 |
| Networks | Burstormer | 37.06/0.9344/0.151 | 0.2399/0.2390 | 29.99/0.8617/0.300 | 0.3549/0.3060 |
| | RBSR | 39.10/0.9498/0.117 | 0.2074/0.2341 | 30.49/0.8713/0.275 | 0.3425/0.2895 |
| | AHDRNet | 36.68/0.9279/0.158 | 0.2010/0.2259 | 29.86/0.8589/0.308 | 0.3382/0.2909 |
| HDR | HDRGAN | 35.94/0.9177/0.181 | 0.1995/0.2178 | 30.00/0.8592/0.337 | 0.3555/**0.3109** |
| Reconstruction | HDR-Tran. | 37.62/0.9356/0.129 | 0.2043/0.2142 | 30.18/0.8662/0.279 | 0.3245/0.2933 |
| Networks | SCTNet | 37.47/0.9443/0.122 | 0.2348/0.2260 | 30.13/0.8644/0.281 | 0.3415/0.2936 |
| | Kim *et al.* | 39.09/0.9494/0.115 | 0.2467/0.2388 | 30.28/0.8658/**0.268** | 0.3302/0.2954 |
| Our TMRNet | w/o Ada. | **39.35/0.9516/0.112** | 0.2003/0.2181 | **30.65/0.8725/**0.270 | 0.3422/0.2898 |
| | w/ Ada. | - | **0.2537/0.2422** | - | **0.3676**/0.3020 |

metrics are not completely stable and are only used for auxiliary evaluation. The actual visual results can better demonstrate the effect of different methods. As shown in Fig. 4, applying other models trained on synthetic data to the real world easily produces undesirable artifacts. Benefiting from the proposed self-supervised real-image adaptation, our results have fewer artifacts and more satisfactory content. More visual comparisons can be seen in Appendix J.

**Inference Time.** Our method has a similar inference time with RBSR (Wu et al., 2023), and a shorter time than recent state-of-the-art ones, *i.e.*, BIPNet (Dudhane et al., 2022), Burstormer (Dudhane et al., 2023), HDR-Tran. (Liu et al., 2022), SCTNet (Tel et al., 2023) and Kim *et al.* (Kim & Kim, 2023). Overall, our method maintains good efficiency while improving performance compared to recent state-of-the-art methods. Detailed comparisons can be seen in Appendix B.

# 6 ABLATION STUDY

## 6.1 EFFECT OF NUMBER OF INPUT FRAMES

To validate the effect of the number of input frames, we conduct experiments by removing relatively higher exposure frames one by one, as shown in Tab. 3. Naturally, more frames result in better performance. In addition, adding images with longer exposure will lead to exponential increases of shooting time. The higher the exposure time, the less valuable content in the image. Considering these two aspects, we only adopt 5 frames. Furthermore, we conduct experiments with more combinations of multi-exposure images in Appendix G.

## 6.2 EFFECT OF TMRNET

We change the depths of common and specific modules to explore the effect of temporal modulation in TMRNet. For a fair comparison, we keep the total depth the same. From Tab. 4, completely

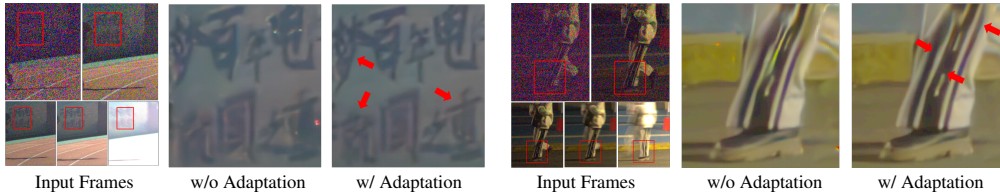

| Input Frames | w/o Adaptation | w/ Adaptation | Input Frames | w/o Adaptation | w/ Adaptation |

Figure 5: Effect of self-supervised real-image adaptation. Our results have fewer ghosting artifacts and more details in the areas indicated by the red arrow. Please zoom in for better observation.

Table 3: Effect of number of input multi-exposure frames.

| Input | BracketIRE PSNR↑/SSIM↑/LPIPS↓ | BracketIRE+ PSNR↑/SSIM↑/LPIPS↓ |
|---|---|---|
| $\mathbf{Y}_1$ | 29.64/0.8235/0.340 | 25.13/0.7289/0.466 |
| $\{\mathbf{Y}_i\}_{i=1}^2$ | 33.93/0.8923/0.234 | 27.99/0.8003/0.390 |
| $\{\mathbf{Y}_i\}_{i=1}^3$ | 36.98/0.9294/0.165 | 29.70/0.8446/0.324 |
| $\{\mathbf{Y}_i\}_{i=1}^4$ | 38.70/0.9460/0.127 | 30.41/0.8645/0.286 |
| $\{\mathbf{Y}_i\}_{i=1}^5$ | **39.35/0.9516/0.112** | **30.65/0.8725/0.270** |

Table 4: Effect of number (*i.e.*, $a_c$ and $a_s$) of common and specific blocks.

| $\alpha_c$ | $\alpha_s$ | BracketIRE PSNR↑/SSIM↑/LPIPS↓ | BracketIRE+ PSNR↑/SSIM↑/LPIPS↓ |
|---|---|---|---|
| 0 | 40 | 38.96/0.9491/0.120 | 30.41/0.8700/0.276 |
| 8 | 32 | 39.26/0.9512/0.115 | **30.70**/0.8721/**0.270** |
| 16 | 24 | **39.35/0.9516/0.112** | 30.65/**0.8725/0.270** |
| 24 | 16 | 39.10/0.9497/0.117 | 30.59/0.8713/0.271 |
| 32 | 8 | 39.16/0.9500/0.117 | 30.59/0.8722/0.275 |
| 40 | 0 | 39.10/0.9498/0.117 | 30.49/0.8713/0.275 |

taking common modules or specific ones does not achieve satisfactory results, as the former ignores the degradation difference of multi-exposure images while the latter may be difficult to optimize. Allocating appropriate depths to both modules can perform better. In addition, we also conduct experiments by changing the depths of the two modules independently in Appendix H.

## 6.3 Effect of Self-Supervised Adaptation

We regard TMRNet trained on synthetic pairs as a baseline to validate the effectiveness of the proposed adaptation method on BracketIRE task. From the visual comparisons in Fig. 5, the adaptation method reduces artifacts significantly and enhances some details. From the quantitative metrics, it improves CLIPIQA (Wang et al., 2023a) and MANIQA (Yang et al., 2022) from 0.2003 and 0.2181 to 0.2537 and 0.2422, respectively. Please kindly refer to Appendix I for more results.

## 7 Conclusion

Existing multi-image processing methods typically focus exclusively on either restoration or enhancement, which are insufficient for obtaining visually appealing images with clear content in low-light conditions. Motivated by the complementary potential of multi-exposure images in denoising, deblurring, HDR reconstruction, and SR, we utilize exposure bracketing photography to combine these tasks to get a high-quality image. Specifically, we suggested a solution that initially pre-trains the model with synthetic pairs and subsequently adapts it to unlabeled real-world images, where a temporally modulated recurrent network and a self-supervised adaptation method are presented. Moreover, we constructed a data simulation pipeline for synthesizing pairs and collected real-world images from 200 nighttime scenarios. Experiments on both datasets show our method achieves better results than state-of-the-arts.

## 8 Applications and Limitations

**Applications.** A significant application of this work is HDR imaging at night, especially in dynamic environments, aiming to obtain noise-free, blur-free, and HDR images. Such images can clearly show both bright and dark details in nighttime scenes. The application is not only challenging but also practically valuable. We also experiment with it on a smartphone (*i.e.*, Xiaomi 10S), as shown in Figs. G and H.

**Limitations.** Given the diverse imaging characteristics (especially noise model parameters) of various sensors, our method necessitates tailored training for each sensor. In other words, our model trained on images from one sensor may exhibit limited generalization ability when applied to other sensors. We leave the investigation of a more general model for future work.

## ACKNOWLEDGMENTS

This work was supported by the National Key R&D Program of China (2022YFA1004100) and the National Natural Science Foundation of China (NSFC) under Grant U22B2035.

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

# APPENDIX

The content of the appendix involves:

- More implementation details in Appendix A.
- Comparison of computational costs in Appendix B.
- Results on other datasets in Appendix C.
- Comparison with all-in-one methods in Appendix D.
- Comparison with step-by-step processing in Appendix E.
- Comparison with burst imaging in Appendix F.
- Effect of multi-exposure combinations in Appendix G.
- Effect of TMRNet in Appendix H.
- Effect of self-supervised adaptation in Appendix I.
- More visual comparisons in Appendix J.

## A  IMPLEMENTATION DETAILS

### A.1  RGB-TO-RAW CONVERSION IN DATA SIMULATION PIPELINE

We provide an illustration to visually demonstrate the pipeline of synthesizing data in Fig. A.

The original HDR videos (Froehlich et al., 2014) in the synthetic dataset are shot by the Alexa camera, a CMOS sensor based motion picture camera made by Arri. UPI (Brooks et al., 2019) is used to convert these RGB videos to raw space. Note that we do not use the same camera parameters as UPI (Brooks et al., 2019) setting, but use the parameters from Alexa camera during RGB-to-RAW conversion.

### A.2  FRAME INTERPOLATION IN DATA SIMULATION PIPELINE

RIFE (Huang et al., 2022) is used to interpolate between two frames, and it does not affect the data distribution. From visual observation of interpolation results, it also supports this point. Nonetheless, limited the ability of RIFE, the synthetic blur still has a gap with the real-world one. In this work, the proposed self-supervised real-image adaptation method is to alleviate this gap. Besides, recent interpolation models are more capable of dealing with large motion (Jain et al., 2024) and complex textures (Zhong et al., 2024). We believe this problem can also be alleviated with the advancement of interpolation models.

### A.3  NOISE IN DATA SIMULATION PIPELINE

The noise in raw images is mainly composed of shot and read noise (Brooks et al., 2019). Shot noise can be modeled as a Poisson random variable whose mean is the true light intensity measured in photoelectrons. Read noise can be approximated as a Gaussian random variable with a zero mean and a fixed variance. The combination of shot and read noise can be approximated as a single heteroscedastic Gaussian random variable $\mathbf{N}$, which can be written as,

$$\mathbf{N} \sim \mathcal{N}(\mathbf{0}, \lambda_{read} + \lambda_{shot}\mathbf{X}), \tag{A}$$

where $\mathbf{X}$ is the clean signal value. $\lambda_{read}$ and $\lambda_{shot}$ are determined by sensor's analog and digital gains.

In order to make our synthetic noise as close as possible to the collected real-world image noise, we adopt noise parameters of the main camera sensor in Xiaomi 10S smartphone, and they (*i.e.*, $\lambda_{shot}$

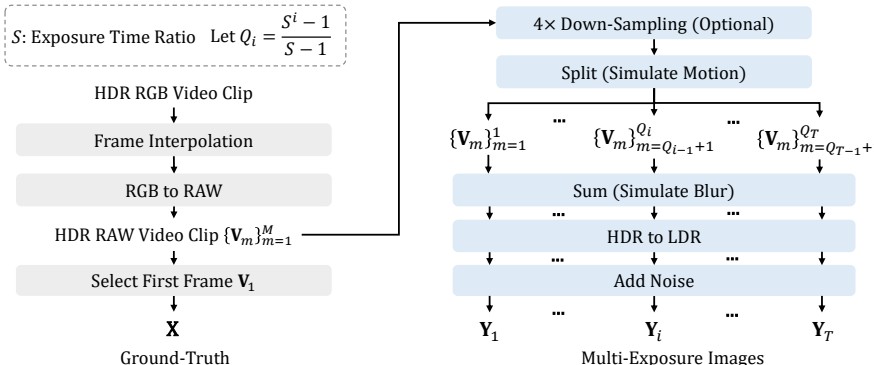

Figure A: Overview of data simulation pipeline. We utilize HDR video to synthesize multi-exposure images $\{\mathbf{Y}_i\}_{i=1}^{T}$ and the corresponding GT image $\mathbf{X}$. $S$ denotes the exposure time ratio between $\mathbf{Y}_i$ and $\mathbf{Y}_{i-1}$. $\mathbf{Y}_i$ is obtained by summing and processing $S^{i-1}$ ($\{i-1\}$-th power of $S$) images from HDR raw video $\mathbf{V}$. $Q_i$ denotes the total number of images from $\mathbf{V}$ that participate in constructing $\{\mathbf{Y}_k\}_{k=1}^{i}$.

and $\lambda_{read}$) can be found in the metadata of raw image file. Specifically, the ISO of all captured real-world images is set to 1,600. At this ISO, $\lambda_{shot} \approx 2.42 \times 10^{-3}$ and $\lambda_{read} \approx 1.79 \times 10^{-5}$. Moreover, in order to synthesize noise with various levels, we uniformly sample the parameters from ISO = 800 to ISO = 3,200. Finally, $\lambda_{read}$ and $\lambda_{shot}$ can be expressed as,

$$\log(\lambda_{shot}) \sim \mathcal{U}(\log(0.0012), \log(0.0048)),$$
$$\log(\lambda_{read}) \mid \log(\lambda_{shot}) \sim \mathcal{N}(1.869\log(\lambda_{shot}) + 0.3276, 0.3^2), \tag{B}$$

where $\mathcal{U}(a, b)$ represents a uniform distribution within the interval $[a, b]$.

### A.4 EVALUATION SETTINGS

For synthetic experiment evaluation, 10 and 4 invalid pixels around the original input image are excluded for BracketIRE and BracketIRE+ tasks, respectively. Here is the reason. The surrounding 10 pixels of the original HDR videos[2] (Froehlich et al., 2014) are all 0 values. Thus, the surrounding 10 and 4 pixels of the synthetic input image are all 0 values for BracketIRE and BracketIRE+ task, respectively. In practice, this situation hardly occurs, so we exclude them for evaluation. Nonetheless, the model can deal with marginal areas. Actually, in the early version of our paper, we used all pixels for evaluation. After the suggestions from peers, we changed the evaluation way, as the current way is more in line with the actual situation.

## B COMPARISON OF COMPUTATIONAL COSTS

We provide comparisons of the inference time, as well as the number of FLOPs and model parameters in Tab. A. We suggest inference time as the main reference for computational cost comparison, as the testing time is more significant than the number of FLOPs and model parameters in practical applications. It can be seen that our method has a similar time with RBSR (Wu et al., 2023), and a shorter time than recent state-of-the-art ones, *i.e.*, BIPNet (Dudhane et al., 2022), Burstormer (Dudhane et al., 2023), HDR-Tran. (Liu et al., 2022), SCTNet (Tel et al., 2023) and Kim *et al.* (Kim & Kim, 2023). Overall, our method maintains good efficiency while improving performance compared to recent state-of-the-art methods.

Further, in the recent state-of-the-art methods, BIPNet (Dudhane et al., 2022), RBSR (Wu et al., 2023), and our TMRNet are based on the convolutional neural network (CNN), while Burstormer (Dudhane et al., 2023), HDR-Tran. (Liu et al., 2022), SCTNet (Tel et al., 2023), and Kim *et al.* (Kim & Kim, 2023) are based on Transformer. TMRNet has similar #FLOPs to RBSR and lower #FLOPs than BIPNet, but higher #FLOPs than these Transformer-based methods. We think this is acceptable and

---

[2]https://www.hdm-stuttgart.de/vmlab/hdm-hdr-2014/.

Table A: Comparison of #parameters and computational costs with state-of-the-art methods when generating a $1920 \times 1080$ raw image on BracketIRE task. Note that the inference time can better illustrate the method's efficiency than #parameters and #FLOPs for practical applicability.

| Method | #Params (M) | #FLOPs (G) | Time (ms) |
|---|---|---|---|
| DBSR (Bhat et al., 2021a) | 12.90 | 16,120 | 850 |
| MFIR (Bhat et al., 2021b) | 12.03 | 18,927 | 974 |
| BIPNet (Dudhane et al., 2022) | 6.28 | 135,641 | 6,166 |
| Burstormer (Dudhane et al., 2023) | 3.11 | 9,200 | 2,357 |
| RBSR (Wu et al., 2023) | 5.64 | 19,440 | 1,467 |
| AHDRNet (Yan et al., 2019) | 2.04 | 2,053 | 208 |
| HDRGAN (Niu et al., 2021) | 9.77 | 2,410 | 158 |
| HDR-Tran. (Liu et al., 2022) | 1.69 | 1,710 | 1,897 |
| SCTNet (Tel et al., 2023) | 5.02 | 5,145 | 3,894 |
| Kim *et al.* (Kim & Kim, 2023) | 22.74 | 5,068 | 1,672 |
| TMRNet | 13.29 | 20,040 | 1,425 |

Table B: Results on Kalantari *et al.* (Kalantari et al., 2017) dataset for HDR image reconstruction.

| Method | PSNR↑ / SSIM↑ |
|---|---|
| AHDRNett (Yan et al., 2019) | 41.14 / 0.9702 |
| HDRGAN (Niu et al., 2021) | 41.57 / 0.9865 |
| HDR-Tran. (Liu et al., 2022) | 42.18 / 0.9884 |
| SCTNet (Tel et al., 2023) | 42.29 / 0.9887 |
| Kim *et al.* (Kim & Kim, 2023) | 41.99 / 0.9890 |
| TMRNet | 42.43 / 0.9893 |

understandable for two reasons. First, although Transformer-based methods have an advantage in #FLOPs, they bring higher inference time and it is more difficult for them to deploy into embedded chips for practical application. Second, the main idea of TMRNet is to assign specific parameters for each frame while sharing some parameters. We implement this idea based on a more recent CNN-based method (*i.e.*, RBSR), and the basic modules only adopt simple residual blocks. For TMRNet, the basic modules can be easily replaced with #FLOPs-friendly modules, which has great potential for #FLOPs reduction. We plan to experiment with it and provide a lightweight TMRNet in the next version.

## C    RESULTS ON OTHER DATASETS

### C.1    EFFECTIVENESS OF TMRNET ON OTHER DATASETS

To evaluate the effectiveness of TMRNet on other datasets, we conduct experiments on Kalantari *et al*. (Kalantari et al., 2017) dataset for HDR image reconstruction. We compare our TMRNet with recent HDR reconstruction methods. The results in Tab. B show that TMRNet achieves the best results. We also conduct experiments on BurstSR (Bhat et al., 2021a) dataset for burst image super-resolution. We compare our TMRNet with recent burst super-resolution methods. The results in Tab. C show that TMRNet still achieves the best results.

### C.2    EFFECTIVENESS OF SELF-SUPERVISED REAL-IMAGE ADAPTATION ON OTHER DATASETS

To evaluate the effectiveness of self-supervised real-image adaptation on other datasets, we conduct an experiment on burst image super-resolution dataset (Bhat et al., 2021a), which is the only commonly used multi-image processing dataset with both synthetic and real-world data. We first pre-train our TMRNet on synthetic data, and then use our self-supervised loss to fine-tune it on real-world training dataset. The results in Tab. D show that our self-supervised loss brings 0.87 dB PSNR gain on real-world testing dataset, demonstrating its effectiveness.

Table C: Results on BurstSR (Bhat et al., 2021a) dataset for burst image super-resolution.

| Method | PSNR↑ / SSIM↑ |
|---|---|
| DRSR (Bhat et al., 2021a) | 48.05 / 0.984 |
| MFIR (Bhat et al., 2021b) | 48.33 / 0.985 |
| BIPNet (Dudhane et al., 2022) | 48.49 / 0.985 |
| Burstormer (Dudhane et al., 2023) | 48.82 / 0.986 |
| RBSR (Wu et al., 2023) | 48.80 / 0.987 |
| TMRNet | 48.92 / 0.987 |

Table D: Effectiveness of self-supervised real-image adaptation on BurstSR (Bhat et al., 2021a) dataset for burst image super-resolution.

| Method | PSNR↑ / SSIM↑ |
|---|---|
| w/o Self-Supervised Adaptation | 44.70 / 0.9690 |
| w/ Self-Supervised Adaptation | 45.57 / 0.9734 |

## D    COMPARISON WITH ALL-IN-ONE METHODS

All-in-one models (Zhang et al., 2024a; Cui et al., 2024) mean that the models can process images with different degradations. There are three main differences between them and our work. First, all-in-one models generally input a single image, and output a single image. Our model inputs multi-exposure images, and outputs a single image. Second, in all-in-one models, the degradation type across input samples can be different. In our model, the degradation across input samples is basically consistent, and the degradation is different between multiple images within an input. Third, all-in-one models utilize the capabilities of the model itself to achieve multiple tasks. Our model can additionally exploit the complementarity of the input images to achieve multiple tasks.

We have conducted an experiment using AdaIR (Cui et al., 2024). The original AdaIR model can only input one image. For a fair comparison, we concatenate multi-exposure images aligned by an optical flow network (Ranjan & Black, 2017) together as the input of AdaIR. Its PSNR result is 38.06 dB, which is lower than our 39.35 dB. We argue that the main reason for AdaIR's poor performance is that it is not specifically designed for multi-image processing. In contrast, the methods compared in Tab. 2 are all specific multi-image processing methods.

## E    COMPARISON WITH STEP-BY-STEP PROCESSING

Here we demonstrate our advantages by conducting an ablation study that compares the joint processing and progressive processing manners on BracketIRE task on synthetic dataset. We mark our multi-task joint processing way as 'Denoising&Deblurring&HDR'. In the ablation study, we decompose the whole task with three steps: (1) first denoising, (2) then deblurring, and (3) finally HDR reconstruction. We mark the way as 'Denoising+Deblurring+HDR'. During training, we construct data pairs and modify our TMRNet as the specialized network for each step. The inputs of each step are all multi-exposure images concatenated together, which aim to exploit the complementarity of multi-exposure images in denoising, deblurring, and HDR reconstruction task, respectively. During inference, we sequentially cascade the networks at all steps to test. The results are shown in the Tab. E. It can be seen that step-by-step processing is inferior to joint processing.

Actually, during step-by-step processing, the denoising model performs well. The main reason for the unsatisfactory performance is that the deblurring model has a limited effect when dealing with the severe blur in the long-exposure image. It prevents the HDR reconstruction model from working well. Specifically, in the training phase, the input multi-exposure images of 'HDR' model are blur-free and noise-free. However, in the testing phase, there may still be some blur remaining in the input of 'HDR' model, due to the limited capabilities of 'Deblurring' models. Thus, a data gap between training and testing appears in 'HDR' model, and it hurts the model performance. In contrast, joint processing can avoid this problem.

Table E: Comparison with step-by-step processing.

| Manner | PSNR↑ / SSIM↑ / LPIPS↓ |
|---|---|
| Step-by-Step Processing (3 Steps, Denoising+Deblurring+HDR) | 37.93 / 0.9367 / 0.120 |
| Our Joint Processing (1 Step, Denoising&Deblurring&HDR) | 39.35 / 0.9516 / 0.112 |

Table F: Comparison with burst processing manner. $\{\mathbf{Y}_i^b\}_{b=1}^5$ denote the 5 burst images with exposure time $\Delta t_i$.

| Input | BracketIRE PSNR↑ / SSIM↑ / LPIPS↓ | BracketIRE+ PSNR↑ / SSIM↑ / LPIPS↓ |
|---|---|---|
| $\{\mathbf{Y}_1^b\}_{b=1}^5$ | 32.22 / 0.8606 / 0.271 | 26.89 / 0.7663 / 0.416 |
| $\{\mathbf{Y}_2^b\}_{b=1}^5$ | 35.05 / 0.9237 / 0.171 | 28.93 / 0.8289 / 0.345 |
| $\{\mathbf{Y}_3^b\}_{b=1}^5$ | 31.75 / 0.9284 / 0.144 | 28.24 / 0.8581 / 0.302 |
| $\{\mathbf{Y}_4^b\}_{b=1}^5$ | 26.30 / 0.8853 / 0.215 | 24.46 / 0.8225 / 0.381 |
| $\{\mathbf{Y}_5^b\}_{b=1}^5$ | 20.04 / 0.8247 / 0.364 | 20.59 / 0.8062 / 0.450 |
| $\{\mathbf{Y}_i\}_{i=1}^5$ | **39.35 / 0.9516 / 0.112** | **30.65 / 0.8725 / 0.270** |

Table G: Effect of multi-exposure image combinations on BracketIRE task.

| Input | PSNR ↑ / SSIM↑ / LPIPS↓ |
|---|---|
| $\{\mathbf{Y}_1, \mathbf{Y}_2, \mathbf{Y}_3\}$ | 36.98 / 0.9294 / 0.165 |
| $\{\mathbf{Y}_1, \mathbf{Y}_3, \mathbf{Y}_5\}$ | 37.54 / 0.9388 / 0.146 |
| $\{\mathbf{Y}_2, \mathbf{Y}_3, \mathbf{Y}_4\}$ | 36.48 / 0.9463 / 0.127 |
| $\{\mathbf{Y}_3, \mathbf{Y}_4, \mathbf{Y}_5\}$ | 31.31 / 0.9291 / 0.164 |
| $\{\mathbf{Y}_1, \mathbf{Y}_2, \mathbf{Y}_3, \mathbf{Y}_4\}$ | 38.70 / 0.9460 / 0.127 |
| $\{\mathbf{Y}_2, \mathbf{Y}_3, \mathbf{Y}_4, \mathbf{Y}_5\}$ | 36.54 / 0.9483 / 0.122 |
| $\{\mathbf{Y}_1, \mathbf{Y}_2, \mathbf{Y}_3, \mathbf{Y}_4, \mathbf{Y}_5\}$ | **39.35 / 0.9516 / 0.112** |

Besides, joint processing only produces one model for deploying. It can simplify the complexity of the entire imaging system and make it easier to deploy in actual scenarios. Benefiting from this, joint processing way is also being pursued by some mobile phone manufacturers, as far as we know.

## F  COMPARISON WITH BURST IMAGING

To validate the effectiveness of leveraging multi-exposure frames, we compare our method with burst imaging manner that employs multiple images with the same exposure. For each exposure time $\Delta t_i$, we use our data simulation pipeline to construct 5 burst images $\{\mathbf{Y}_i^b\}_{b=1}^5$ as inputs. The quantitative results are shown in Tab. F. It can be seen that the models using moderate exposure bursts (*e.g.*, $\mathbf{Y}_2$ and $\mathbf{Y}_3$) achieve better results, as these bursts take good trade-offs between noise and blur, as well as overexposure and underexposure. Nevertheless, their results are still weaker than ours by a wide margin, mainly due to the limited dynamic range of the input bursts.

## G  EFFECT OF MULTI-EXPOSURE COMBINATIONS

We conduct experiments with different combinations of multi-exposure images in the Tab. G. From Tab. G, the more frames, the better the results. More generally, we argue that as the number of frames increases, the worst case is that the model does not extract useful information from the increased frame. In other words, adding frames does not lead to the worse results, but leads to the similar or better ones. In this work, we adopt the frame number $T = 5$ and exposure time ratio $S = 4$, as it can cover most of the dynamic range using fewer frames. Additionally, without considering shooting and computational costs, it is foreseeable that a larger $T$ or smaller $S$ would perform better when keeping the overall dynamic range the same.

Table H: Effect of depth of specific blocks while keeping common blocks the same on BracketIRE task. $a_c$ and $a_s$ denote the number of common and specific blocks, respectively.

| $\alpha_c$ | $\alpha_s$ | Time (ms) | PSNR↑ / SSIM↑ / LPIPS↓ |
|---|---|---|---|
| 16 | 0 | 808 | 38.66 / 0.9462 / 0.125 |
| 16 | 8 | 1,016 | 38.87 / 0.9480 / 0.122 |
| 16 | 16 | 1,224 | 39.12 / 0.9496 / 0.116 |
| 16 | 24 | 1,425 | 39.35 / 0.9516 / **0.112** |
| 16 | 32 | 1,633 | **39.36** / **0.9518** / 0.114 |

Table I: Effect of depth of common blocks while keeping specific blocks the same on BracketIRE task. $a_c$ and $a_s$ denote the number of common and specific blocks, respectively.

| $\alpha_c$ | $\alpha_s$ | Time (ms) | PSNR↑ / SSIM↑ / LPIPS↓ |
|---|---|---|---|
| 0 | 24 | 1,015 | 38.91 / 0.9484 / 0.121 |
| 8 | 24 | 1,219 | 39.15 / 0.9502 / 0.117 |
| 16 | 24 | 1,425 | **39.35** / **0.9516** / **0.112** |
| 24 | 24 | 1,637 | 39.31 / 0.9512 / 0.115 |

Table J: Effect of loss terms for self-supervised real-image adaptation. '-' denotes TMRNet trained on synthetic pairs. 'NaN' implies the training collapse. Note that the no-reference metrics are not completely stable and are provided only for auxiliary evaluation.

| $\mathcal{L}_{ema}$ | $\mathcal{L}_{self}$ | BracketIRE CLIPIQA↑ / MANIQA↑ | BracketIRE+ CLIPIQA↑ / MANIQA↑ |
|---|---|---|---|
| - | - | 0.2003 / 0.2181 | 0.3422 / 0.2898 |
| ✓ | ✗ | 0.2003 / 0.2181 | 0.3422 / 0.2898 |
| ✗ | ✓ | NaN / NaN | NaN / NaN |
| ✓ | $\lambda_{self} = 0.5$ | 0.2295 / 0.2360 | 0.3591 / 0.2978 |
| ✓ | $\lambda_{self} = 1$ | **0.2537** / 0.2422 | 0.3676 / 0.3020 |
| ✓ | $\lambda_{self} = 2$ | 0.2270 / 0.2391 | **0.3815** / 0.3172 |
| ✓ | $\lambda_{self} = 4$ | 0.1974 / **0.2525** | 0.3460 / **0.3189** |

## H    EFFECT OF TMRNET

For TMRNet, we conduct experiments by changing the depths of the common and specific blocks independently, whose results are shown in Tab. H and Tab. I, respectively. Denote the number of common and specific blocks by $a_c$ and $a_s$, respectively. On the basis of $a_c = 16$ and $a_s = 24$, adding their depths did not bring significant improvement while increasing the inference time. We speculate that it could be attributed to the difficulty of optimization for deeper recurrent networks.

## I    EFFECT OF SELF-SUPERVISED ADAPTATION

In the main text, we regard TMRNet trained on synthetic pairs as a baseline to validate the effectiveness of the proposed adaptation method. Here we provide more visual comparisons in Fig. B, where our results have fewer speckling and ghosting artifacts, as well as more details both in static and dynamic scenes.

We also provide the quantitative comparisons in Tab. J. It can be seen that the proposed adaptation method can bring both CLIPIQA (Wang et al., 2023a) and MANIQA (Yang et al., 2022) improvements. In addition, only deploying $\mathcal{L}_{ema}$ would make the network parameters to be not updated. Without $\mathcal{L}_{ema}$, the self-supervised fine-tuning would lead to a trivial solution, thus collapsing. This is because the result of inputting all frames is not subject to any constraints at this time.

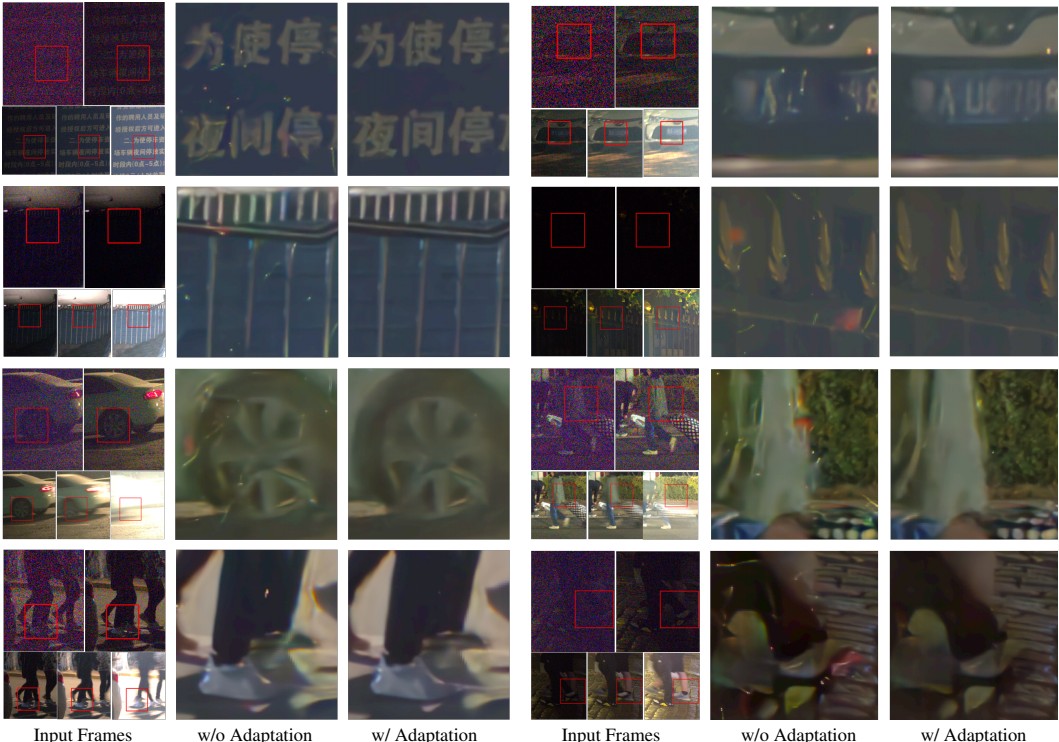

Input Frames    w/o Adaptation    w/ Adaptation    Input Frames    w/o Adaptation    w/ Adaptation

Figure B: Effect of self-supervised real-image adaptation. Our results have fewer speckling and ghosting artifacts, as well as more details. The top four show the visual effects of static scenes (but with camera motion), and the bottom four show the visual effects of moving objects. Please zoom in for better observation.

Moreover, we empirically adjust the weight $\lambda_{self}$ of $\mathcal{L}_{self}$ and conduct experiments with different $\lambda_{self}$. From Tab. J, the effect of $\lambda_{self}$ on the results is acceptable. It is worth noting that although higher $\lambda_{self}$ (e.g., $\lambda_{self} = 2$ and $\lambda_{self} = 4$) sometimes achieves higher quantitative metrics, the image contrast decreases and the visual effect is unsatisfactory at this time. This also demonstrates the no-reference metrics are not completely stable, thus we only take them for auxiliary evaluation. Focusing on the visual effects, we set $\lambda_{self} = 1$.

## J  MORE VISUAL COMPARISONS

We first provide more visual comparisons on BracketIRE+ task. Figs. C and D show the qualitative comparisons on the synthetic images. Figs. E and F show the qualitative comparisons on the real-world images. It can be seen that our method generates more photo-realistic images with fewer artifacts than others.

Moreover, in order to observe the effect of dynamic range enhancement, we provide some full-image results from real-world dataset. Note that the size of the original full images is very large, and here we downsample them for display. Fig. G shows the full-image visualization results on BracketIRE task. Fig. H shows the full-image visualization results on BracketIRE+ task. Our results preserve both bright and dark details, showing a higher dynamic range.

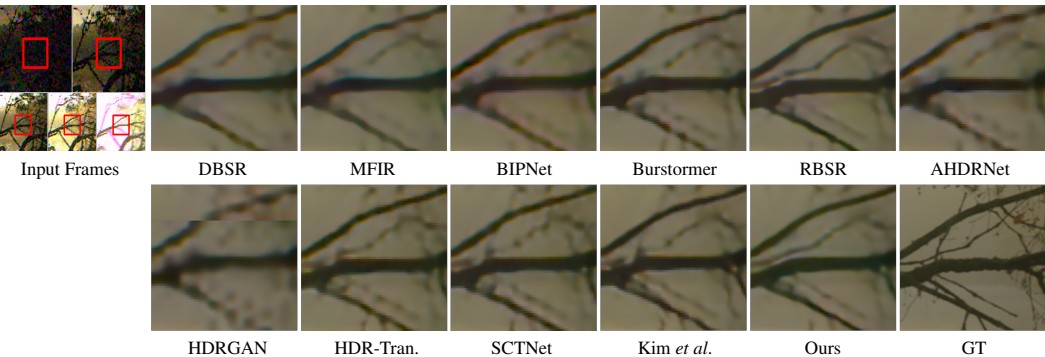

Figure C: Visual comparison on the synthetic dataset of BracketIRE+ task. Our result restores clearer details. Please zoom in for better observation.

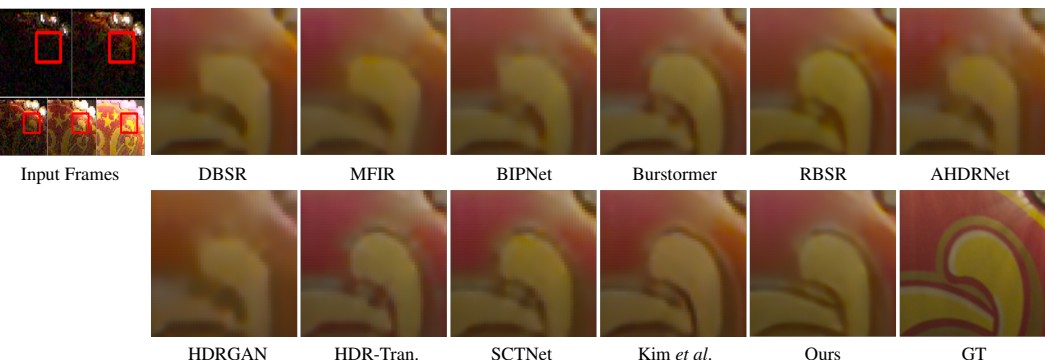

Figure D: Visual comparison on the synthetic dataset of BracketIRE+ task. Our result restores more fidelity content. Please zoom in for better observation.

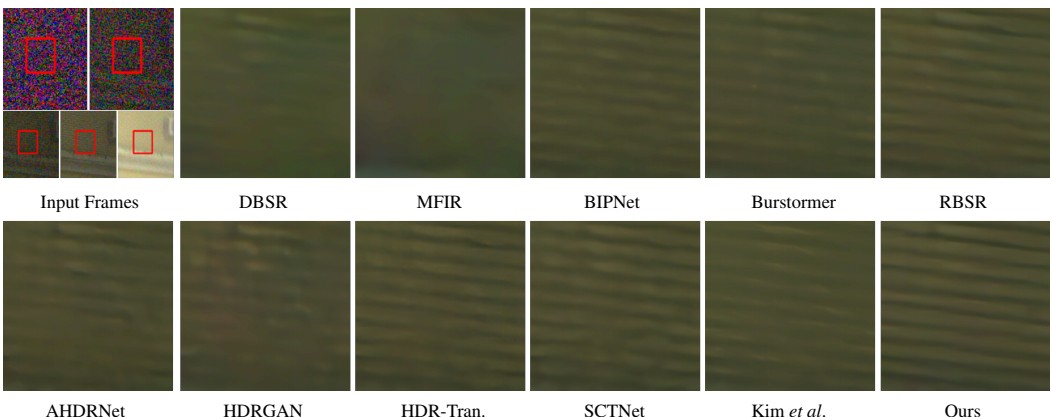

Figure E: Visual comparison on the real-world dataset of BracketIRE+ task. Our result restores clearer textures. Please zoom in for better observation.

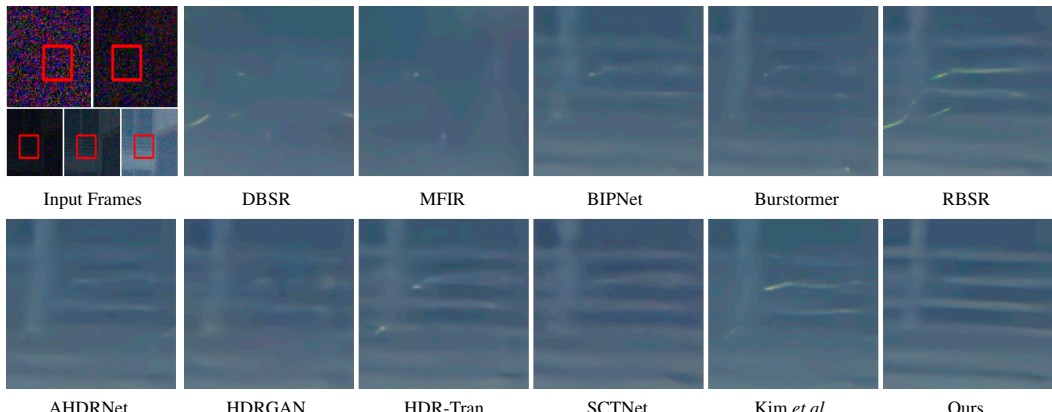

Figure F: Visual comparison on the real-world dataset of BracketIRE+ task. Our result has fewer artifacts. Please zoom in for better observation.

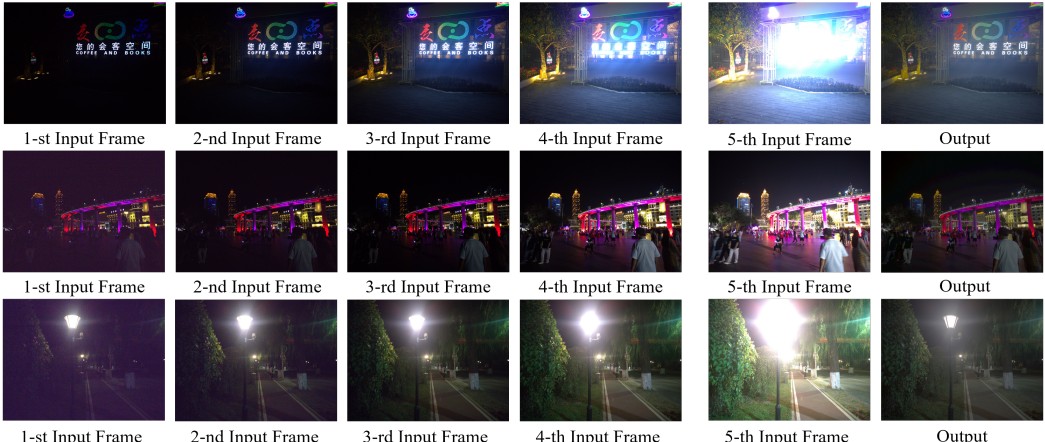

Figure G: Full-image results on the real-world dataset of BracketIRE task. Our results preserve both the bright areas in short-exposure images and the dark areas in long-exposure images. Note that the size of the original full images is very large, and here we downsample them for display. Please zoom in for better observation. Moreover, we provide a higher resolution version in README.md of the supplementary material.

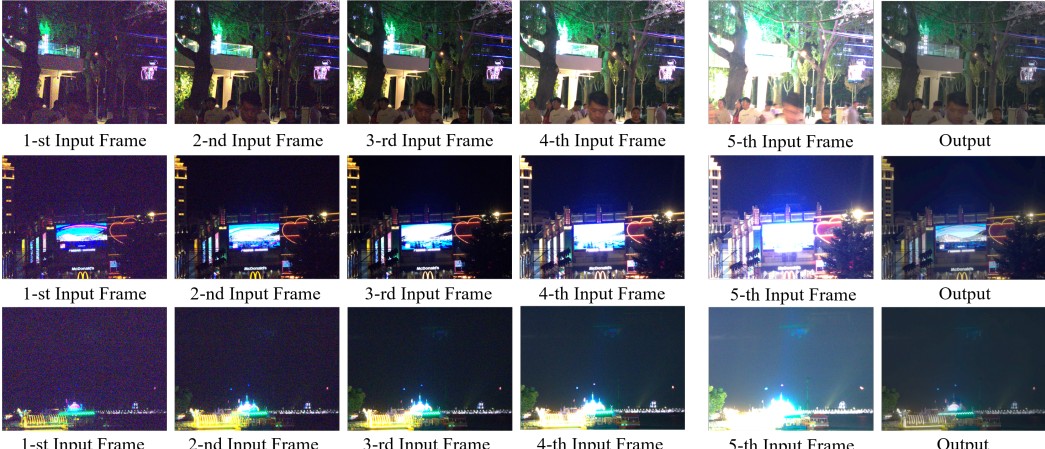

Figure H: Full-image results on the real-world dataset of BracketIRE+ task. Our results preserve both the bright areas in short-exposure images and the dark areas in long-exposure images. Note that the size of the original full images is very large, and here we downsample them for display. Please zoom in for better observation. Moreover, we provide a higher resolution version in README.md of the supplementary material.

