# OpenReview forum: "Exposure Bracketing Is All You Need For A High-Quality Image"
_ICLR.cc/2025/Conference — ICLR 2025 Poster_

### Official Review · Reviewer_MkdG · 2024-10-16

**Soundness:** 4
**Presentation:** 4
**Contribution:** 3
**Rating:** 8
**Confidence:** 5

**Summary:**

This paper addresses the challenge of acquiring high-quality photos in low-light environments by unifying image restoration and enhancement tasks using exposure bracketing photography. This paper proposes a solution that includes a temporally modulated recurrent network (TMRNet) and a self-supervised adaptation method, pre-training the model on synthetic paired data and then adapting it to real-world unlabeled images. They also construct a data simulation pipeline and collect real-world images from diverse nighttime scenarios. Experiments show that their method outperforms state-of-the-art multi-image processing techniques.

**Strengths:**

This paper presents a paradigm for addressing low-light imaging through exposure bracketing photography with a simple yet effective network structure. This practical technical approach is orthogonal to mainstream research, which can more easily inspire subsequent studies.
The core contribution of this paper lies in the construction of the dataset and the corresponding training methods, particularly the self-supervised adaptation method. The dataset creation method is solid, and the design of the self-supervised adaptation method is clever.
The paper is well-organized, with clearly articulated problems, solutions, and experimental results that are easy to understand.
The impressive experiments show clear advantages over the compared methods.

**Weaknesses:**

1. **The use of RIFE**

   In previous work simulating realistic blur on the UPI-based REDS video deblurring dataset, I tried using RIFE to interpolate frames to 1920fps. While RIFE is an advanced frame interpolation algorithm, I observed that it still introduces artifacts in areas with significant motion and complex textures. These limitations may reduce the generalization ability of models trained on such synthetic data when applied to real-world blurred scenes.

2. **Comparison with AI-ISP**

   While the proposed self-supervised adaptation method shows promise, it still faces challenges when paired data is available compared to supervised methods. It would be helpful if the authors could clarify how this approach compares to AI-ISP, which addresses RAW denoising, super-resolution, deblurring, and HDR processing as independent subtasks. By decomposing these tasks, it is possible (though challenging) to obtain paired real data to replace synthetic data, potentially enhancing performance in practice. Alternatively, the authors could demonstrate that exposure bracketing provides a clear advantage over handling each subtask independently on synthetic data as an indirect validation of its benefits.

Addressing these concerns would clarify the practical advantages of the proposed approach, and I would be inclined to improve my rating accordingly.

**Questions:**

I think the flowchart of the synthetic paired dataset is critical, would it be better to move it from the supplementary material to the manuscript?

[After Rebuttal]: The authors have addressed my concerns, thus I am inclined to improve my rating.

---

> ### Author Response · Authors · 2024-11-21
> **Response to Reviewer MkdG**
>
> We thank you for your valuable comments and suggestions. We appreciate your questions and hope our responses could address your concerns.
>
>
> 1. Question: The use of RIFE
>
> Response: We agree with your point that the synthetic blur still has a gap with the real-world one due to the limitations of the frame interpolation model. In this work, the proposed self-supervised real-image adaptation method is to alleviate this gap. Besides, recent interpolation models are more capable of dealing with large motion [1] and complex textures [2]. We believe this problem will also be alleviated with the advancement of interpolation models. We have added them in Appendix A.2.
>
> [1] Jain, Siddhant and Watson, Daniel and Tabellion, Eric and Poole, Ben and Kontkanen, Janne and others. Video interpolation with diffusion models. In CVPR, 2024.
>
> [2] Zhong, Zhihang and Krishnan, Gurunandan and Sun, Xiao and Qiao, Yu and Ma, Sizhuo and Wang, Jian. Clearer Frames, Anytime: Resolving Velocity Ambiguity in Video Frame Interpolation. In ECCV, 2024.
>
> 2. Question: Comparison between self-supervised and supervised manners
>
> Response: For supervised methods, large amounts of high-quality paired data are critical to model performance. However, it is very expensive or even impossible to obtain a large amount of paired data. For self-supervised methods, since ground-truth is not required, the available data is much larger. Benefiting from this, self-supervised methods may further enhance performance by expanding the data scale. And with the improvement of self-supervised algorithms, we believe that they can approach or reach the performance of supervised methods. Thus, it is necessary and valuable to explore self-supervised methods in the current era.
>
>
> 3. Question: Comparison with AI-ISP
>
> Response: As far as we know, there are almost no suitable open-source AI-ISPs for comparison. Here we indirectly demonstrate our advantages by conducting an ablation study that compares the joint processing and progressive processing manners on BracketIRE task on synthetic dataset.
>
>
> We mark our multi-task joint processing way as "Denoising\&Deblurring\&HDR". In the ablation study, we decompose the whole task with three steps: (1) first denoising, (2) then deblurring, and (3) finally HDR reconstruction. We mark the way as "Denoising + Deblurring + HDR". During training, we construct data pairs and modify our TMRNet as the specialized network for each step. The inputs of each step are all multi-exposure images concatenated together, which aim to exploit the complementarity of multi-exposure images in denoising, deblurring, and HDR reconstruction task, respectively.  During inference, we sequentially cascade the networks at all steps to test. The results are shown in the following table. It can be seen that step-by-step processing is inferior to joint processing.
>
> Actually, during step-by-step processing, the denoising model performs well. The main reason for the unsatisfactory performance is that the deblurring model has a limited effect when dealing with the severe blur in the long-exposure image. It prevents the HDR reconstruction model from working well. Specifically, in the training phase, the input multi-exposure images of "HDR" model are blur-free and noise-free. However, in the testing phase, there may still be some blur remaining in the input of "HDR" model, due to the limited capabilities of "Deblurring" models. Thus, a data gap between training and testing appears in "HDR" model, and it hurts the model performance. In contrast, joint processing can avoid this problem.
>
> Besides, joint processing only produces one model for deploying. It can simplify the complexity of the entire imaging system and make it easier to deploy in actual scenarios. Benefiting from this, the way of joint processing is also being pursued by some mobile phone manufacturers, as far as we know.
>
> We have added them in Appendix E.
>
> | Manner | PSNR/SSIM/LPIPS | Computational cost
> | :------:       | :------:  |  :------:
> | Step-by-step processing (3 steps, denoising + deblurring + HDR)       |  37.93/0.9367/0.120    |  x 3
> | Our joint processing (1 step, denoising\&deblurring\&HDR)     | 39.35/0.9516/0.112    |  x 1
>
>
> 4. Question: Would it be better to move the flowchart of the synthetic paired dataset from the supplementary material to the manuscript?
>
> Response: Yes, we agree with that. Due to space limitations, we moved it to the supplementary material when we submitted the review version. In the camera-ready version, we will try to move it back to the main text.

---

> ### Comment · Reviewer_MkdG · 2024-11-28
>
> I sincerely appreciate the authors' response.
>
> (Discussion:) I believe that simulating realistic motion blur using frame interpolation can only serve as a temporary solution. Modeling continuous motion through inherently inaccurate discrete data has inevitable limitations. It might be more reasonable to generate blur kernels based on motion estimation rather than directly relying on frame interpolation. Perhaps data production schemes based on computer graphics are more promising for the future.
>
> By the way, I find the academic community's current focus on works modifying data pipelines somewhat strange. When it comes to deploying raw image processing tasks, the limited computational resources and runtime memory of edge devices (e.g., smartphones) mean that almost no frameworks proposed in published research can be directly applied in practice. Therefore, the real question should always be: *What effective information increment does this work bring to the community?*—not **just** *What novel ideas does this work include?* I have encountered far too many low-level vision papers with "novel" ideas that ultimately fail to provide practical value, serving little purpose beyond helping students fulfill graduation requirements.
>
> For a practical application-oriented study like this, I believe the most significant contribution lies in demonstrating the advantages of exposure bracketing for multi-task scenarios and offering a robust, reproducible framework. In my view, this is the strongest aspect of the paper and the main reason I am inclined to rate it highly.
>
> Overall, my experience in the industry allows me to resonate with the authors' perspective. Therefore, I am leaning towards keeping the original rating.

---

> > ### Author Response · Authors · 2024-11-28
> >
> > We sincerely thank you for your detailed and insightful feedback. We are particularly encouraged by your inclination to rate this work highly. Below, we address some key points from your discussion.
> >
> > First, for the motion blur synthesis, we agree with your view that data production schemes based on computer graphics are more promising for the future. More recent work [1] explored this idea. We believe this idea will become increasingly feasible and effective as 3D and 4D scene reconstruction techniques continue to advance.
> >
> > Second, We deeply resonate with your observation about the challenges of deploying low-level vision frameworks on edge devices. In my own low-level vision research experience, I have also faced this problem, struggling between novelty and practicality. In the end, I realized that practicality may be the main thing, novelty may be only the icing on the cake, and application-oriented novelty is worth exploring. In this work, we explore novel ideas (i.e., TMRNet and self-supervised real-image adaptation) while being practical (i.e., leveraging exposure bracketing to get a high-quality image).
> >
> > Third, we are particularly encouraged by your acknowledgment of the advantages of exposure bracketing in multi-task scenarios and your appreciation of the robustness and reproducibility of our framework. We are thrilled they resonated with you.
> >
> > Once again, we sincerely thank you for your thoughtful feedback and recognition. Your insights motivate us to continue pursuing impactful and practical research.
> >
> > [1] GS-Blur: A 3D Scene-Based Dataset for Realistic Image Deblurring. NeurIPS 2024. https://arxiv.org/abs/2410.23658

---

### Official Review · Reviewer_QkhS · 2024-10-28

**Soundness:** 3
**Presentation:** 4
**Contribution:** 3
**Rating:** 5
**Confidence:** 5

**Summary:**

The authors proposed a neural network capable of performing denoising, deblurring, HDR, and SR altogether using exposure bracketing. The proposed network structure has a novel feature, called temporally modulated recurrent network, which is claimed to be effective in handling multiple images with different degradation. To train the network, the authors also constructed a dataset by applying synthetic degradation sequentially. The two loss functions are also developed to train the network in a self-supervised manner.

**Strengths:**

1. The proposed method is the first method that performs the four restoration and enhancement tasks altogether.
2. The necessity of the temporal modulated recurrent network is clearly explained, and its effectiveness is supported by experimental results.
3. The necessity of the introduced loss functions is clearly explained, and their effectiveness is supported by experimental results.
4. The constructed synthetic and real datasets will be helpful for researchers in the related field.
5. The paper is well-written and well-organized.

**Weaknesses:**

1. I understand that ICLR accepts application-oriented papers. However, from the theoretical point-of-view, the manuscript does not contain sufficient technical novelties. The network can perform the four different tasks altogether since it is simply trained using the synthetic (and real) dataset containing four degradation. In other words, except for the dataset, the network does not have significant novelties compared to existing video restoration networks.

2. The proposed temporal modulation network seems not very original. This reviewer cannot see a significant difference between the proposed network and a commonly used conv-LSTM (with different conv layers for each layer). Several other temporal modulation networks have also been explored, e.g., Temporal Modulation Network for Controllable Space-Time Video Super-Resolution, CVPR 2021.

3. The proposed self-supervised loss functions are also not very different from existing loss terms, e.g., Exponential Moving Average Normalization for Self-supervised and Semi-supervised Learning, CVPR 2021.

4. The effectiveness of the proposed network module and loss functions needs to be justified by other public datasets.

**Questions:**

Since the paper is especially well-written, I do not have specific questions for clarification. See my concerns in the Weakness section.

---

> ### Author Response · Authors · 2024-11-21
> **Response to Reviewer QkhS**
>
> We thank you for your valuable comments and suggestions. We appreciate your questions and hope our responses could address your concerns.
>
>
> 1. Question: Problem setting and motivation
>
> Response: We kindly remind the reviewers that our work does not simply train a model using data containing four degradations. It is problem-oriented and has clear motivations.
>
>
> Our work aims to process multiple images with different exposures to generate a single high-quality image.  Multi-exposure images, especially those taken in low-light environments, inevitably possess noise, blur, and low dynamic range. Meanwhile, the magnitude and parameters of these degradations vary across multi-exposure images, making the multi-exposure images complementary in removing each degradation. We propose to leverage the complementarity of multi-exposure images to integrate the four problems (i.e., denoising, deblurring, high dynamic range reconstruction, and super-resolution) into a unified framework, generating a noise-free, blur-free, high dynamic range (HDR), and high-resolution image. To the best of our knowledge, this is the first work to achieve multi-exposure image processing from this perspective. Additionally, compared with previous multi-image processing work, the degradations we consider are more comprehensive and more consistent with the actual situation.
>
> The work itself is not directly oriented to application, but expects to utilize the complementarity of multi-exposure images on multiple tasks to solve multiple problems together. Even in terms of application, HDR imaging at night is only a single significant one. Our work also has the potential to be applied in low-light denoising and motion deblurring.
>
>
>
> 2. Question: Novelty of the proposed TMRNet
>
> Response: Conv-LSTM and the existing temporal sequence processing networks (e.g., RBSR we compared) generally use the same module parameters when processing different sequences, since these sequences are of similar type (e.g., degradation). In our work, TMRNet suggests learning different parameters for each frame sequence, as frames in multi-exposure images suffer different degradations. In Figure 1, modules with different colors have different parameters. Compared to the baseline network (i.e., RBSR) in Figure 1 (a), TMRNet can achieve a 0.25dB PSNR improvement.
>
> Our TMRNet is similar to [1] only in naming, while the contributions and main focus are completely different. In [1], 'temporal modulation' means to modulate the feature according to time \textit{t}, thus interpolating an intermediate frame corresponding to \textit{t}. However, the module parameters in [1] that process different frames are still shared. In our TMRNet, 'temporal modulation' means to learn specific parameters for each frame, i.e., the module parameters that process different frames are not fully shared. We also take [1] to experiment on our synthetic dataset, and its PSNR result is 36.14 dB, which is far behind our outcome of 39.35 dB.
>
> In general, TMRNet is problem-oriented with clear motivation, and has novelty.
>
> [1] Temporal Modulation Network for Controllable Space-Time Video Super-Resolution. CVPR 2021.
>
>
> 3. Question: Novelty of the proposed self-supervised loss
>
>
> Response: Our self-supervised real-image adaptation loss contains two terms, i.e., temporally self-supervised loss and EMA regularization loss. The former plays a major role and is significantly different from [2]; the latter plays a supporting role and can be considered similar to [2]. As shown in Figure 2 (a) and Equation (8), temporally self-supervised loss uses results with more input frames to supervise results with fewer input frames. This idea exploits the temporal characteristics of multi-exposure image processing, i.e., the more input frames, the better the result. As far as we know, this idea has not appeared in related works and is novel.
>
>
> [2] Exponential Moving Average Normalization for Self-supervised and Semi-supervised Learning. CVPR 2021.

---

> ### Author Response · Authors · 2024-11-21
> **Response to Reviewer QkhS**
>
> 4. Question: Effectiveness of TMRNet on other datasets
>
> Response: We conduct experiments on Kalantari [3] dataset for HDR image reconstruction. We compare our TMRNet with recent HDR reconstruction methods. The following results show that TMRNet achieves the best results.
>
> | Method       |  PSNR / SSIM
> |:---:         |:---:
> | AHDRNet      | 41.14 / 0.9702
> | HDRGAN       | 41.57 / 0.9865
> | HDR-Tran.    | 42.18 / 0.9884
> | SCTNet       | 42.29 / 0.9887
> | Kim et al.   | 41.99 / 0.9890
> | Our TMRNet   | 42.43 / 0.9893
>
>
> We also conduct experiments on BurstSR [4] dataset for burst image super-resolution. We compare our TMRNet with recent burst super-resolution methods. The following results show that TMRNet still achieves the best results.
>
>
> | Method     |  PSNR / SSIM
> |:---:       |:---:
> | DRSR       | 48.05 / 0.984
> | MFIR       | 48.33 / 0.985
> | BIPNet     | 48.49 / 0.985
> | Burstormer | 48.82 / 0.986
> | RBSR       | 48.80 / 0.987
> | Our TMRNet | 48.92 / 0.987
>
> We have added them in Appendix C.
>
>
>
> [3] Deep High Dynamic Range Imaging of Dynamic Scenes. ACM TOG 2017.
>
> [4] Deep Burst Super-Resolution. CVPR 2021.
>
>
> 5. Question: Effectiveness of self-supervised loss on other datasets
>
> Response: We conduct an experiment on burst image super-resolution dataset [4], which is the only commonly used multi-image processing dataset with both synthetic and real-world data. We first pre-train our TMRNet on synthetic data, and then use our self-supervised loss to fine-tune it on real-world training dataset. The following results show that our self-supervised loss brings 0.87 dB PSNR gain on real-world testing dataset, demonstrating its effectiveness.
>
> | Method                    |  PSNR / SSIM
> | :---:                     | :---:
> | w/o self-supervised loss  | 44.70 / 0.9690
> | w/ self-supervised loss   | 45.57 / 0.9734
>
> We have added it in Appendix C.

---

> ### Author Response · Authors · 2024-12-01
>
> Dear Reviewer QkhS,
>
> We sincerely appreciate the time and effort you have dedicated to reviewing our paper. With the discussion period deadline approaching (in 42 hours), we kindly request your feedback if there are any remaining concerns or suggestions you would like us to address.
>
> Thank you once again for your valuable contributions to enhancing the quality of our work.
>
> Authors of Submission 4958

---

### Official Review · Reviewer_GQoP · 2024-11-02

**Soundness:** 3
**Presentation:** 3
**Contribution:** 2
**Rating:** 6
**Confidence:** 5

**Summary:**

This paper proposes a temporally modulated recurrent network based on exposure bracketing photography for image restoration and enhancement in low-light environments. This paper simulates the paired data and pre-trains the temporally modulated recurrent network with synthetic images. Additionally, a self-supervised adaptation method is utilized to improve the model's robustness in real-world unlabeled images. Experiments presented in the paper validate its effectiveness for multi-image processing.

**Strengths:**

1. This work is well-complete overall, including data simulation, real-world data collection, and method design and experiments.
2. The experimental results show improvements in both visual quality and quantitative metrics, demonstrating its effectiveness.

**Weaknesses:**

1. The comparison methods in Table 2 are all trained on synthetic data, while TMRNet is fine-tuned on real-world data. This comparison is somewhat unfair. It is recommended that the authors include the results of TMRNet without fine-tuning on real-world data and provide a note.
2. This paper claims that more frames result in better performance, would it introduce frames with over-exposure or large blur, making the restoration process more challenging?
3. In summarizing the contributions, the authors claim to unify image restoration and enhancement tasks, which seems somewhat exaggerated. First, this is not a new task; in fact, methods of HDR have already been addressing issues such as noise, blur, or ghosting. Additionally, even if the input includes images with various degradations, this has also appeared in previous papers, such as R1 and R2. Therefore, it would be better if the authors could change this contribution point or use a more appropriate wording.

R1. Low-light image restoration with short-and long-exposure raw pairs;
R2. Digital gimbal: End-to-end deep image stabilization with learnable exposure times;

**Questions:**

Refer to Weaknesses.

---

> ### Author Response · Authors · 2024-11-21
> **Response to Reviewer GQoP**
>
> We thank you for your valuable comments and suggestions. We appreciate your questions and hope our responses could address your concerns.
>
>
> 1. Question: Add the results without fine-tuning TMRNet on real-world data to Table 2
>
> Response: Thanks for the suggestion. The first line in Table J shows the results on real-world data without fine-tuning TMRNet. We have added them in Table 2 and provided a note in the revision.
>
>
> 2. Question: Would introducing more frames with over-exposure or large blur make the restoration process more challenging?
>
> Response: 'More frames result in better performance' is just a conclusion for Table 3, which is the case at least within 5 frames. Moreover, in our data, the frame with the highest exposure is already highly blurry and overexposed, as shown in the lower left side of Figure 4.
>
> More generally, we think that as the number of frames increases, the worst case is that the network does not extract useful information from the increased frame. In other words, adding frames does not lead to the worse results, but leads to the similar or better ones. For each scene in our dataset, we have compared the results when taking the varying number of frames as input. As frames increase, the results do not worsen in almost all scenes. This observation supports our view to some extent.
>
> We have added these in Appendix G.
>
> 3. Question: Task setting, degradation setting, and contribution point statement
>
>
> Response: The main difference between our works and previous works is that previous works only achieved limited tasks (see Table 1), and did not bring out the potential of multi-exposure images. For example, [1] and [2] only consider denoising and deblurring tasks, but do not involve HDR and super-resolution tasks. Our work points out the complementarity of multi-exposure images on multiple tasks (i.e., denoising, deblurring, HDR reconstruction, and SR), and leverages this to jointly address these tasks.
>
>
> Moreover, most HDR reconstruction works are only verified on clean and sharp inputs, i.e., they ignore noise and blur degradations in multi-exposure images. Although recent HDR reconstruction works have considered noise, almost no work has considered blur in long-exposure images as far as we know. Our work takes degradation factors into account more realistically and comprehensively, incorporating both noise and blur degradation. Furthermore, even in [1] and [2] that focus on denoising and deblurring tasks (but do not involve HDR reconstruction), they only consider blur from camera shake, and do not consider blur from object movement. Our degradation contains both camera motion blur and object motion blur (see the first line in Figure H).
>
> For the first contribution point statement, we have realized that the 'unify image restoration and enhancement tasks' statement may be somewhat exaggerated. We have changed it to 'We propose to utilize exposure bracketing photography to get a high-quality (i.e., noise-free, blur-free, high dynamic range, and high-resolution) image by combining image denoising, deblurring, high dynamic range reconstruction, and super-resolution tasks.', which may be more in line with the practical contribution and application of this work. The title and the related statements in the paper have also been revised.
>
>
> [1] Low-light image restoration with short- and long-exposure raw pairs.
>
> [2] Digital gimbal: End-to-end deep image stabilization with learnable exposure times.

---

> ### Author Response · Authors · 2024-12-01
>
> Dear Reviewer GQoP,
>
> We sincerely appreciate the time and effort you have dedicated to reviewing our paper. With the discussion period deadline approaching (in 42 hours), we kindly request your feedback if there are any remaining concerns or suggestions you would like us to address.
>
> Thank you once again for your valuable contributions to enhancing the quality of our work.
>
> Authors of Submission 4958

---

### Official Review · Reviewer_jyRM · 2024-11-04

**Soundness:** 3
**Presentation:** 3
**Contribution:** 3
**Rating:** 5
**Confidence:** 4

**Summary:**

The authors propose a framework for a new setting of image restoration. It uses raw images as input and process several kinds degradation in one network. The architecture can perform well.

**Strengths:**

1. A novel setting combing several image restoration tasks, utilizing raw space images.
2. The performance is good enough.
3. Extensive experiments are done to prove characteristics of the proposed method.

**Weaknesses:**

1. The BracketIRE and BracketIRE+ are similar with all-in-one. The authors should discuss on this issue.
2. The computational cost seems to be high. The fairness of comparison is doubtful.
3. The datasets must be publicly, otherwise the work is not so essential.

**Questions:**

1. The way to convert RGB into raw is not suitable. For the method used in UPI, it need parameters from cameras. What is the type of CMOS for those videos? Is it the same with the setting of UPI? Would the RIFE affect data distribution of the original videos?
2. As the input is raw images, do the authors perform packing operation for those data? Many methods would do packing to split color channels.
3. Why do the authors  exclude 10 and 4 invalid pixels around the original input image? Do the proposed method deal with marginal areas?
4. The FLOPs of TMRNet is huge compared with others.
5. It seems to be the first work using this kind of unified image restoration setting. For the BracketIRE, can it compare with some all-in-one methods, such as AdaIR and Perceive-IR?
6. Why do the authors add SR for BracketIRE+? To match the title as restoration tasks or to show the ability on low resolution images?

Scores can be raised upon replies.

---

> ### Author Response · Authors · 2024-11-21
> **Response to Reviewer jyRM**
>
> We thank you for your valuable comments and suggestions. We appreciate your questions and hope our responses could address your concerns.
>
> 1. Question: Comparision with all-in-one methods
>
>
> Response: All-in-one models mean that the models can process images with different degradations. There are three main differences between them and our work. First, all-in-one models generally input a single image, and output a single image. Our model inputs multi-exposure images, and outputs a single image. Second, in all-in-one models, the degradation type across input samples can be different. In our model, the degradation across input samples is basically consistent, and the degradation is different between multiple images within an input. Third, all-in-one models utilize the capabilities of the model itself to achieve multiple tasks. Our model can additionally exploit the complementarity of the input images to achieve multiple tasks.
>
> Perceive-IR is not publicly available, and we have conducted an experiment using AdaIR. The original AdaIR model can only input one image. For a fair comparison, we concatenate multi-exposure images aligned by an optical flow network together as the input of AdaIR. Its PSNR result is 38.06 dB, which is lower than our 39.35 dB. We argue that the main reason for AdaIR’s poor performance is that it is not specifically designed for multi-image processing. In contrast, the methods compared in Table 2 are all specific multi-image processing methods.
>
>
> 2. Question: Computational cost
>
> Response: We suggest inference time as the main reference for computational cost comparison, as the testing time is more important in practical applications. Our method has a similar inference time with RBSR, and a shorter time than recent state-of-the-art ones, i.e., BIPNet, Burstormer, HDR-Transformer, SCTNet, and Kim et al.
>
> In the recent state-of-the-art methods, BIPNet, RBSR, and our TMRNet are based on the convolutional neural network (CNN), and Burstormer, HDR-Transformer, SCTNet, and Kim et al. are based on Transformer. TMRNet has similar \#FLOPs to RBSR and lower \#FLOPs than BIPNet, while TMRNet has higher \#FLOPs than these Transformer-based methods. We think this is acceptable and understandable for two reasons. First, although Transformer-based methods have an advantage in \#FLOPs, they bring higher inference time and it is more difficult for them to deploy into embedded chips for practical application. Second, the main idea of TMRNet is to assign specific parameters for each frame while sharing some parameters. We implement this idea based on a more recent CNN-based method (i.e., RBSR), and the basic modules only adopt simple residual blocks. For TMRNet, the basic modules can be easily replaced with \#FLOPs-friendly modules, which has great potential for \#FLOPs reduction. We plan to experiment with it and provide a lightweight TMRNet in the next version.
>
> We will add this in the revision.
>
> 3. Question: What is the type of CMOS for those videos? Is it the same with the setting of UPI? Would the RIFE affect data distribution of the original videos?
>
>
> Response: The original HDR videos [1] in the synthetic dataset are shot by the Alexa camera, a CMOS sensor based motion picture camera made by Arri. We do not use the same parameter setting with UPI, but use the parameter setting from Alexa camera to convert RGB into raw images. We will add the details in the revision and make the data synthesis code public after the paper is accepted.
>
>
> Moreover, RIFE is only used to interpolate between two frames, and it does not affect the data distribution. From visual observation of interpolation results, it also supports this point.
>
>
> [1] Froehlich, Jan, et al. "Creating cinematic wide gamut HDR-video for the evaluation of tone mapping operators and HDR-displays." Digital photography X. Vol. 9023. SPIE, 2014.
>
>
> 4. Question: Do the authors perform packing operation for raw data?
>
> Response: Yes, we perform a packing operation for the raw data before feeding it into the model. We will add the details in the revision.
>
> 5. Question: Why do the authors exclude 10 and 4 invalid pixels around the original input image? Do the proposed method deal with marginal areas?
>
> Response: The surrounding 10 pixels of the original HDR video [1] are all 0 values. Thus, the surrounding 10 and 4 pixels of the synthetic input image are all 0 values for BracketIRE and BracketIRE+ task, respectively. In practice, this situation hardly occurs, so we exclude them for evaluation.
>
> The model can deal with marginal areas. Actually, in the early version of our paper, we used all pixels for evaluation. After the suggestions from peers, we changed the evaluation way, as the current way is more in line with the actual situation.

---

> ### Author Response · Authors · 2024-11-21
> **Response to Reviewer jyRM**
>
> 6. Question: Why do the authors add SR for BracketIRE+? To match the title as restoration tasks or to show the ability on low resolution images?
>
> Response: The main purpose of our work is to explore the complementarity and usability of multi-exposure images in performing multiple tasks. According to [2], the sub-pixel shift between multiple images caused by camera shake or motion is conducive to multi-frame super-resolution. Thus, we add super-resolution task for BracketIRE+, showing the ability of multi-exposure images in super-resolution task.
>
> In addition, in practical applications, it is possible to demand higher-resolution images while performing noise-free, blur-free, and HDR imaging. The task setting of BracketIRE+ is also to meet this actual need.
>
>
> [2] Wronski, Bartlomiej, et al. "Handheld multi-frame super-resolution." ACM Transactions on Graphics (ToG) 38.4 (2019): 1-18.
>
> 7. Question: Are code and dataset publicly available?
>
> Response: The main code, pre-trained models, and all datasets are already publicly available. Due to the double-blind policy, we did not add them to the paper. We will add this link to the camera-ready version.

---

> > ### Comment · Reviewer_jyRM · 2024-11-26
> >
> > The authors have solved my concerns. However, the revision paper is not uploaded by the authors. I would recommend a revised version for better evaluation.

---

> > > ### Author Response · Authors · 2024-12-01
> > >
> > > Dear Reviewer jyRM,
> > >
> > > We sincerely appreciate the time and effort you have dedicated to reviewing our paper. With the discussion period deadline approaching (in 42 hours), we kindly request your additional feedback on the revised manuscript. Does the revision meet the criteria for improving your rating?
> > >
> > > Thank you once again for your valuable contributions to enhancing the quality of our work.
> > >
> > > Authors of Submission 4958

---

> ### Author Response · Authors · 2024-11-28
>
> We sincerely appreciate your feedback and thoughtful suggestions. Addressing your concerns has been an honor and a valuable opportunity to improve our work.
>
> We have revised the paper thoroughly.
> * Question 1 was added to Appendix D.
> * Question 2 was added to Appendix B.
> * Question 3 was added to Appendix A.1 and Appendix A.2.
> * Question 4 was added to Section 5.1.
> * Question 5 was added to Appendix A.4.
> * The responses for Question 6 and Question 7 were marked with blue in Introduction (Lines 94-96) and Abstract  (Lines 25-27) , respectively.
> * Moreover, the suggested revisions from other reviewers were also added to the paper.
>
> We hope the revised paper can sufficiently address your concerns and provide the necessary support for an improved rating. Let me know if you'd like further refinement or adjustments.

---

### Author Response · Authors · 2024-11-28
**Brief Summary of Paper Revisions**

We sincerely thank all the reviewers for their valuable suggestions. We have revised the paper based on these comments. A brief summary is as follows.

* In Appendix A, we added more implementation details, covering the data simulation pipeline and evaluation settings.
* In Appendix B, we added more analysis on computational cost.
* In Appendix C, we added results of TMRNet and self-supervised real-image adaptation on other datasets.
* In Appendix D, we added a comparison with all-in-one methods.
* In Appendix E, we added a comparison with a step-by-step processing manner.
* In Appendix G, we added more effect analysis on multi-exposure image combinations.
* In the main paper, we revised the first contribution point and related statements to make it clearer.
* Besides, we clarified some statements and added some minor details, as suggested by the reviewers.

We hope the revised paper can effectively address reviewers' concerns. Please let us know if there are any further problems or suggestions.

---

### Meta-Review · Area_Chair_ozsR · 2024-12-19

**Metareview:**

This paper explore exposure bracketing photography to combine several image restoration tasks for high-quality image restoration. Experimental results show the effectiveness of the proposed method.

The paper received reviews with mixed ratings. The major concerns of reviewers include the complexity of the proposed TMRNet, unfair experimental comparisons, and limited technical novelties.

Based on the rebuttal and discussions, the authors solve the most concerns of reviewers. The paper can be accepted.

**Additional Comments On Reviewer Discussion:**

In the discussion stage, the authors solve the concerns of reviewers. However, the authors are suggested to revise the expressions to better describe their major contributions. In addition, the technical novelties should be better clarified clearly.

---

### Decision · Program_Chairs · 2025-01-22

Accept (Poster)